# Interference-Isolated Elastic Weight Consolidation and Knowledge Calibration for Incremental Object Detection

**De Cheng**[1], **Mingyue Zeng**[1], **Zhipeng Xu**[1], **Di Xu**[2]*, **Nannan Wang**[1]*, **Xinbo Gao**[1]
[1]Xidian University, [2]Huawei Technologies Co. Ltd
dcheng@xidian.edu.cn,    zengmingyue@stu.xidian.edu.cn

## Abstract

Incremental Object Detection (IOD) enables AI systems to continuously learn new object classes over time while retaining knowledge of previously learned categories. This capability is essential for adapting to dynamic environments without forgetting prior information. Although existing IOD methods have made progress in mitigating catastrophic forgetting, they usually lack explicit and quantitative modeling of information conflicts during knowledge preservation, making task boundaries ambiguous. Such conflicts often stem from the fact that a single image can contain objects belonging to previous, present, and future tasks, where unlabeled past and future objects are often mistakenly treated as background. In this paper, we propose a novel approach grounded in Elastic Weight Consolidation (EWC) to alleviate conflict knowledge preservation caused by task interference. Specifically, we introduce the Interference-Knowledge-Isolated Elastic Weight Consolidation (IKI-EWC) framework for IOD, which leverages the mispredictions of the old detector on new task data to estimate task conflicts and suppresses them at the parameter level. By reformulating the Bayesian posterior of model parameters, we derive a mathematical relationship between previously learned knowledge and interference knowledge, enabling targeted elimination of conflicts during model weight updates. In addition, we also propose a prototype-based knowledge calibration (PKC) mechanism to further preserve old knowledge during the training of the detector's classification head. This method employs a learnable projection layer to compensate semantic drift in old class prototypes, and then jointly trains the classification head using both calibrated prototypes and current task features, thereby mitigating forgetting caused by classifier updates. Extensive experiments on PASCAL VOC and MS-COCO benchmarks demonstrate the effectiveness of the proposed method, outperforming state-of-the-art approaches in various settings.

## 1 Introduction

Object detection Felzenszwalb et al. (2009); Tompson et al. (2015); Tian et al. (2019) is a fundamental and important computer vision task Kirillov et al. (2023); Croitoru et al. (2023); Cheng et al. (2024); Wang et al. (2025a); Xu et al. (2025); He et al. (2025a); Cheng et al. (2025; 2026) that has seen remarkable advancement in recent years. However, static detectors, which learn predefined classes from static datasets, cannot adapt to dynamic settings of real-world scenarios. Incremental Object Detection (IOD) methods, which aim to continually detect new objects without forgetting previously learned categories, have emerged as an exciting and challenging task to address this issue. Unlike humans, who can incrementally acquire knowledge with minimal disruption, deep learning models are prone to overwriting prior information when exposed to new data. This limitation is a key challenge for IOD, where the model must integrate new categories without forgetting the old ones.

Existing Incremental Object Detection (IOD) methods can be categorized into two main types: knowledge-distillation-based methods Hao et al. (2019a); Liu et al. (2020b); Peng et al. (2020); Zhou et al. (2020); Mo et al. (2024) and parameter-regularization-based methods Liu et al. (2020a). The

---

*Corresponding authors.

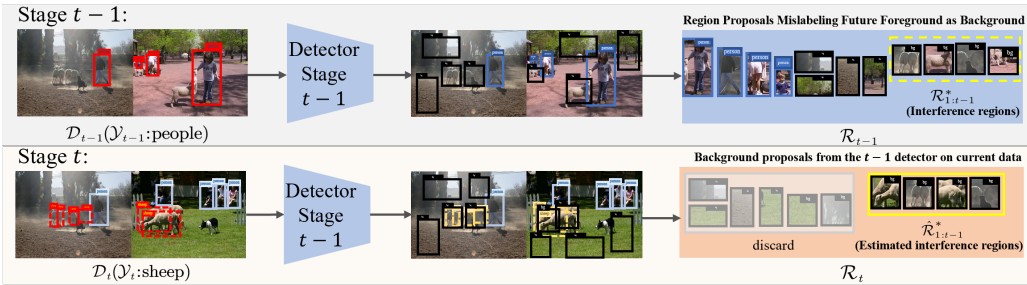

Figure 1: Revisiting the data partitioning between adjacent tasks, the old task data $\mathcal{D}_{t-1}$ contains regions where object classes from the new task $t$ were mistakenly labeled as background, due to the absence of their class labels during training on task $t-1$. We retrospectively estimate the conflicting knowledge using the new task data $\mathcal{D}_t$. The red boxes represent the ground-truth labels, while the dashed red boxes highlight regions that are excluded from the forward pass during model inference.

knowledge-distillation-based approaches typically leverage features or predictions from previous models to transfer knowledge, while parameter-regularization-based approaches impose constraints on model parameters to preserve knowledge from previous tasks. Beyond catastrophic forgetting caused by conventional model parameter updates, IOD poses an additional challenge due to the presence of unlabeled objects from both past and future classes in current training data. These objects are usually misclassified as background, introducing knowledge conflicts between tasks. While the issue of past-class objects can be partially mitigated by generating pseudo labels with the old model, future-class objects remain entirely unlabeled and thus harder to address. This inter-task interference undermines model performance and complicates the stability of incremental learning.

To address these issues, we propose a novel Interference-Knowledge-Isolated Elastic Weight Consolidation and Knowledge Calibration (IIKC) framework for IOD. Unlike existing work Mo et al. (2024), our method avoids the additional cost of training a separate teacher model on the current task data. Instead, we address knowledge conflict from the perspective of parameter regularization. We theoretically build the Interference-Knowledge-Isolated Elastic Weight Consolidation (IKI-EWC) module, bridging the gap between the traditional class incremental learning and IOD tasks on top of the Elastic Weight Consolidation (EWC) Kirkpatrick et al. (2017) method. While the traditional EWC method mitigates catastrophic forgetting by constraining important parameters, our method focuses on isolating interference caused by misclassified past- and future-class objects. Specifically, as shown in Figure 1, at stage $t-1$, the absence of annotations for future objects leads to their misclassification as background. At stage $t$, we estimate this interference by applying the previous detector to current data, and then reformulate the Bayesian parameter estimation to explicitly account for this conflict, enabling effective isolation of task interference. Our proposed method effectively removes the erroneous information (i.e., treating future objects as background) in the parameter importance estimation, thereby alleviating catastrophic forgetting while minimizing the negative impact of knowledge conflicts during new task learning. This Interference-Knowledge-Isolated Elastic Weight Consolidation module is designed to mitigate interference while minimizing its impact on subsequent learning, thereby improving knowledge consolidation for IOD.

In addition, we introduce a novel Prototype-based Knowledge Calibration (PKC) module to further mitigate catastrophic forgetting in the classification head of object detectors. Inspired by Li et al. (2024), PKC addresses semantic drift, which occurs due to inconsistencies between feature representations from old and new tasks. The module employs a trainable projection layer to align the features from previous tasks with those of the current task. This alignment helps preserve the knowledge from old classes while accommodating new ones, ultimately reducing feature drift and alleviating forgetting. By calibrating the prototypes of old classes, PKC improves the model's ability to maintain consistent performance across incremental tasks.

## 2 Related work

**Incremental Learning** techniques can be broadly classified into three categories: memory-based Liu et al. (2021); Luo et al. (2023); Prabhu et al. (2020); Qi et al. (2023), regularization-based Kirkpatrick

et al. (2017); Smith et al. (2021); Sun et al. (2023); Toldo & Ozay (2022), and network architecture-based methods Hu et al. (2023); Wang et al. (2022a;b); Zhou et al. (2022); He et al. (2025b). Memory-based approaches store a small set of samples for replay or generate new samples to compensate for data in subsequent stages. Regularization-based methods aim to stabilize model parameters by controlling feature changes, helping reduce the tendency for forgetting. In practice, this is commonly realized either via knowledge distillation Li & Hoiem (2017); Rebuffi et al. (2017) that aligns logits or intermediate features between the old and new models or via parameter regularization Kirkpatrick et al. (2017); Zenke et al. (2017); Aljundi et al. (2018) that penalizes updates to parameters deemed important for past tasks. Network architecture-based methods dynamically adjust network structures or design specific parameters for each stage to accommodate the changing data stream. In this study, we focus on the parameter regularization approach.

**Incremental Object Detection (IOD)** faces challenges beyond standard class-incremental learning. Besides catastrophic forgetting, unlabeled old or future categories in each stage are often treated as background, creating knowledge conflicts that harm both old and new classes. ILOD Shmelkov et al. (2017) pioneered the knowledge distillation (KD) method for IOD, and many works extended KD to Faster R-CNN to curb forgetting Hao et al. (2019b); Chen et al. (2019); Li et al. (2019). BPF Mo et al. (2024) distills from the combined class probabilities of two teachers to reduce old–new conflicts, and GMDP Wang et al. (2025b) uses Gaussian-mixture prototypes to align feature distributions. In parallel, parameter-regularization methods constrain important weights to stabilize updates Liu et al. (2020a), and rehearsal approaches such as ABR Liu et al. (2023) keep a small class-balanced exemplar set to counter background shift. In this work, we reformulate Bayesian parameter estimation tailored to IOD task, which reduces the impact of knowledge conflicts by incorporating parameter importance. To further mitigate forgetting in classification head, we retrain it using calibrated class prototypes with new task features, where prototypes are calibrated via a trainable projection layer designed to compensate semantic drift between tasks.

## 3 METHOD

### 3.1 PRELIMINARIES

**Problem Definition.** Incremental Object Detection (IOD) focuses on detecting objects over a sequence of learning steps, where each step introduces a new subset of object classes. Let $c$ represent the complete set of object classes to be learned by the detector $\mathcal{M}$ across $T$ steps, such that $c = \bigcup_{i=1}^{T} c_i$ and $c_i \cap c_j = \emptyset$ for all $i \neq j$, where $i, j \in \{1, \ldots, T\}$. Let $\mathcal{D}$ denote the dataset composed of images $\mathbf{x}$ and their corresponding annotations $y$, i.e., $\{(\mathbf{x}, y) \mid (\mathbf{x}, y) \in \mathcal{D}\}$. At step $t$, the model is only provided with annotations for the current class subset $c_t$, even though the images may contain objects from previously seen categories $c_{1:t-1}$. The available annotations $\mathcal{Y}_t$ are thus limited to instances belonging to $c_t$. The key challenge in IOD is to incrementally adapt the model from $\mathcal{M}_{t-1}$ to $\mathcal{M}_t$ by learning the new class subset $c_t$, without access to the previous training data $\{\mathcal{D}_1, \ldots, \mathcal{D}_{t-1}\}$, while still retaining high detection performance on all previously learned classes.

**General Detector Training Pipeline.** We adopt the two-stage Faster R-CNN Ren et al. (2016), comprising a backbone $\mathcal{M}^b$, an RPN $\mathcal{M}^{rpn}$, and a detection head $\mathcal{M}^{head}$. The head pools features on region proposals and uses FC layers to produce region features, which feed a classifier $\boldsymbol{f}^{ch}$ and a box regressor $\boldsymbol{f}^{rh}$. At incremental stage $t$, we write the model as

$$\mathcal{M}_t = \mathcal{M}_t^F \circ \{\boldsymbol{f}_t^{ch}, \boldsymbol{f}_t^{rh}\}, \tag{1}$$

where the region feature extractor $\mathcal{M}_t^F$ includes the backbone, RPN, and the ROI-pooling+FC part of the head (i.e., $\mathcal{M}_t^F = \mathcal{M}_t^b \circ \mathcal{M}_t^{rpn} \circ \text{ROI/FC}$). Here, $\circ$ denotes sequential (left-to-right) composition.

**Elastic Weight Consolidation.** Elastic Weight Consolidation (EWC) Kirkpatrick et al. (2017) is a regularization-based method designed to mitigate catastrophic forgetting in continual learning. The key idea is to preserve knowledge from previous tasks by penalizing large updates to parameters that are deemed important.

From the Bayesian perspective, when data from multiple tasks is learned sequentially, the goal is to maximize the posterior distribution over model parameters given all observed data. Specifically, after observing data from tasks $\mathcal{D}_1$ to $\mathcal{D}_t$, the posterior is written as:

$$p(\boldsymbol{\theta}|\mathcal{D}_{1:t}) \propto p(\mathcal{D}_t|\boldsymbol{\theta}, \mathcal{D}_{1:t-1}) \, p(\boldsymbol{\theta}|\mathcal{D}_{1:t-1}), \tag{2}$$

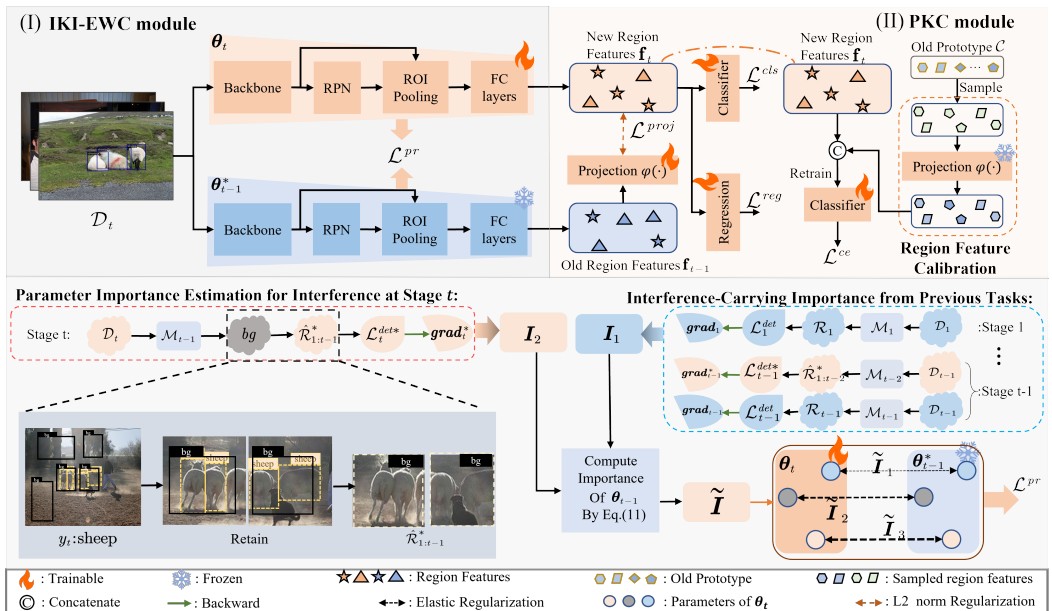

Figure 2: Overview of our proposed IIKC. (I) introduces the **Interference-Knowledge-Isolated Elastic Weight Consolidation (IKI-EWC)** module, which alleviates catastrophic forgetting while resolving knowledge conflicts between old and new tasks. (II) illustrates the **Prototype-based Knowledge Calibration (PKC)** module, which retrains the classification head by combining current task features with old prototypes, effectively preventing catastrophic forgetting. **The lower part** describes the implementation of (I), where the old model $\mathcal{M}_{t-1}$ is used to calculate the importance $\boldsymbol{I}_2$ over the interference region set $\hat{\mathcal{R}}^*_{1:t-1}$. This is then combined with the previous-tasks importance $\boldsymbol{I}_1$ to compute the final parameter importance $\widetilde{\boldsymbol{I}}$ for the old parameters.

In the standard EWC setting, an additional assumption of *conditional independence* is made: $p(\mathcal{D}_t|\boldsymbol{\theta}, \mathcal{D}_{1:t-1}) = p(\mathcal{D}_t|\boldsymbol{\theta})$. This simplifies equation 2 to:

$$p(\boldsymbol{\theta}|\mathcal{D}_{1:t}) \propto p(\mathcal{D}_t|\boldsymbol{\theta})\, p(\boldsymbol{\theta}|\mathcal{D}_{1:t-1}). \tag{3}$$

## 3.2 Overall Framework

To address knowledge conflict and catastrophic forgetting, we propose a novel Interference-Isolated elastic weight consolidation and Knowledge Calibration (IIKC) framework for IOD, as shown in Fig. 2. It contains two main components: **Interference-Knowledge-Isolated Elastic Weight Consolidation (IKI-EWC)** module and **Prototype-based Knowledge Calibration (PKC)** module. First, we use high-confidence predictions from the previous model $\mathcal{M}_{t-1}$ as pseudo-labels to simulate previous task scenarios. This helps eliminate conflict knowledge $\hat{\mathcal{R}}^*_{1:t-1}$ related to previous tasks for estimation of the parameter importance of $\mathcal{M}_{t-1}$, thus improving the retention of the actual important knowledge. Second, to mitigate the severer catastrophic forgetting in the classification head, we introduce a projection layer for knowledge calibration that compensates for semantic drift of the stored old-class prototypes, enabling better integration with current task features during retraining.

## 3.3 Interference-Knowledge-Isolated Elastic Weight Consolidation

We address catastrophic forgetting in IOD task through the lens of parameter regularization. To this end, we revisit the EWC method, which aims to preserve knowledge by constraining updates to important parameters. However, directly applying EWC to IOD raises some issues: due to the absence of future class annotations during earlier-stage model training, the old model $\mathcal{M}_{t-1}$ mistakenly learns to classify new objects as background. EWC then inadvertently preserves this interference knowledge, intensifying the conflict between learning new knowledge and retaining old, ultimately worsening catastrophic forgetting. This interference manifests in the violation of **the conditional independence assumption**, where the factorization from equation 2 to equation 3 no longer holds. To address this, our method, IKI-EWC, identifies and isolates this interference knowledge.

**Rethinking Posterior Factorization for IOD.** In Faster R-CNN, the detection loss $\mathcal{L}^{det}$ comprises the RPN ($\mathcal{L}^{rpn}$) and ROI ($\mathcal{L}^{roi}$) terms, each with a classification loss $\mathcal{L}^{cls}$ and a regression loss $\mathcal{L}^{reg}$:

$$\mathcal{L}^{cls} = \frac{1}{|\mathcal{R}|}\sum_{j\in\mathcal{R}}\mathcal{L}^{ce}(\mathbf{s}_j, y_j), \quad \mathcal{L}^{reg} = \frac{1}{|\mathcal{R}^+|}\sum_{j\in\mathcal{R}^+}\mathrm{SmoothL1}(\mathbf{v}_j - \mathbf{v}_j^{gt}), \tag{4}$$

where $\mathcal{R}$ denotes all region proposals and $\mathcal{R}^+$ the positives. For proposal $j$, $\mathbf{s}_j$ are the classification scores with label $y_j$, and $\mathbf{v}_j, \mathbf{v}_j^{gt}$ are the predicted and target box parameters. Here, $\mathcal{L}^{ce}$ is the cross-entropy loss on class scores, and $\mathrm{SmoothL1}(\cdot)$ is a robust loss for bounding-box regression.

As shown in equation 4, both $\mathcal{L}^{cls}$ and $\mathcal{L}^{reg}$ are computed **per proposal** and then averaged. This naturally motivates a region-based view in which image-level data (with multiple objects) are mapped to proposal-level data for fine-grained analysis. At stage $t$, we treat the current dataset $\mathcal{D}_t$ as a set of proposals $\mathcal{R}_t$ (foreground and background). Proposals accumulated from stages 1 to $t-1$ are denoted $\mathcal{R}_{1:t-1} = \bigcup_{i=1}^{t-1}\mathcal{R}_i$. Let $\mathcal{R}_{1:t-1}^- \subseteq \mathcal{R}_{1:t-1}$ be the background proposals from earlier stages. We define the **interference set** $\mathcal{R}_{1:t-1}^* \subseteq \mathcal{R}_{1:t-1}^-$ as those earlier negatives that actually correspond to stage-$t$ foreground:

$$\mathcal{R}_{1:t-1}^* = \left\{ r \in \mathcal{R}_{1:t-1}^- \ : \ \exists g \in \mathcal{G}_t, \ \mathrm{IoU}(r, g) \geq \gamma \right\}, \tag{5}$$

where $\mathcal{G}_t$ is the set of ground-truth boxes for the new classes at stage $t$ and $\gamma$ is the IoU threshold. These proposals encode *task-conflicting knowledge*: unlabeled in earlier stages, they were learned as background yet now conflict with the stage-$t$ foreground semantics.

We consider tasks $1{:}t$ as our study scope. Following prior work Mo et al. (2024), we assume that pseudo-labeling $\mathcal{D}_t$ with old classes prevents the current proposal set $\mathcal{R}_t$ from containing elements of the past positive set $\mathcal{R}_{1:t-1}^+$. Under this assumption, the data $\mathcal{D}_{1:t}$ decompose as

$$\mathcal{D}_{1:t} = \left(\mathcal{R}_{1:t-1} \setminus \mathcal{R}_{1:t-1}^*\right) \cup \mathcal{R}_t = \mathcal{R}_{1:t-1}' \cup \mathcal{R}_t, \tag{6}$$

where $\mathcal{R}_{1:t-1}' = \mathcal{R}_{1:t-1} \setminus \mathcal{R}_{1:t-1}^*$ collects the non-conflicting proposals from past stages.

*Rederiving the posterior in the IOD setting.* Based on the separation in equation 6, we re-derive the posterior $p(\boldsymbol{\theta} \mid \mathcal{D}_{1:t})$ for IOD. Assuming approximate independence between non-conflicting past proposals and current proposals, the posterior factorizes as

$$p(\boldsymbol{\theta} \mid \mathcal{D}_{1:t}) \ \propto \ p(\mathcal{R}_t \mid \boldsymbol{\theta})\, p(\boldsymbol{\theta} \mid \mathcal{R}_{1:t-1}'), \tag{7}$$

where $p(\mathcal{R}_t \mid \boldsymbol{\theta})$ corresponds to the stage-$t$ detection objective, and $p(\boldsymbol{\theta} \mid \mathcal{R}_{1:t-1}')$ acts as an EWC-style consolidation term built from non-conflicting past proposals.

**IKI-EWC Method in Practice.** For equation 6, $\mathcal{R}_{1:t-1}'$ cannot be computed in isolation during either the $1{:}t-1$ stages or stage $t$. In the former, $\mathcal{G}_t$ is not yet available; in the latter, $\mathcal{R}_{1:t-1}$ is no longer accessible. Importantly, the interfering subset $\mathcal{R}_{1:t-1}^*$ relies on evidence across stages: it needs the background set $\mathcal{R}_{1:t-1}^-$ from $1{:}t-1$ and the labels $\mathcal{G}_t$ at stage $t$ to identify past negatives that are actually new-class foreground. Thus $\mathcal{R}_{1:t-1}^*$ cannot be obtained during $1{:}t-1$ alone. We therefore approximate it at stage $t$ using $\mathcal{D}_t$ and the old model $\mathcal{M}_{t-1}$, and denote the estimate by $\hat{\mathcal{R}}_{1:t-1}^*$.

Concretely, we run $\mathcal{M}_{t-1}$ on $\mathcal{D}_t$ to obtain old-class pseudo labels $\hat{\mathcal{Y}}_{1:t-1}$ (new classes masked), and construct a simulated past set $\hat{\mathcal{D}}_{1:t-1}$ using $\mathcal{D}_t$ images with $\hat{\mathcal{Y}}_{1:t-1}$ as ground truth. Feeding $\hat{\mathcal{D}}_{1:t-1}$ back into $\mathcal{M}_{t-1}$ yields proposals $\hat{\mathcal{R}}_{1:t-1}$ and their background subset $\hat{\mathcal{R}}_{1:t-1}^-$. We then define

$$\hat{\mathcal{R}}_{1:t-1}^* = \left\{ r \in \hat{\mathcal{R}}_{1:t-1}^- \ : \ \exists g \in \mathcal{G}_t, \ \mathrm{IoU}(r, g) \geq \gamma \right\}. \tag{8}$$

At stage $t$, we estimate the interference set $\hat{\mathcal{R}}_{1:t-1}^*$. Because the raw past proposals $\mathcal{R}_{1:t-1}$ are unavailable, $\mathcal{R}_{1:t-1}'$ cannot be formed at the data level. By definition, the non-conflicting part would satisfy

$$\hat{\mathcal{R}}_{1:t-1}' \ = \ \mathcal{R}_{1:t-1} \setminus \hat{\mathcal{R}}_{1:t-1}^*. \tag{9}$$

Viewing $\mathcal{R}_{1:t-1}$ as a mixture of a clean subset $\hat{\mathcal{R}}_{1:t-1}'$ and an interference subset $\hat{\mathcal{R}}_{1:t-1}^*$, and letting $k = \dfrac{|\hat{\mathcal{R}}_{1:t-1}^*|}{|\hat{\mathcal{R}}_{1:t-1}'|}$ denote their relative mass, we can solve for the clean posterior (see Appendix B.1):

$$p(\boldsymbol{\theta} \mid \hat{\mathcal{R}}_{1:t-1}') \ = \ (1+k)\, p(\boldsymbol{\theta} \mid \mathcal{R}_{1:t-1}) \ - \ k\, p(\boldsymbol{\theta} \mid \hat{\mathcal{R}}_{1:t-1}^*). \tag{10}$$

Here $k$ quantifies the severity of interference as the ratio of estimated interfering proposals to non-conflicting ones. The term $p(\boldsymbol{\theta} \mid \mathcal{R}_{1:t-1})$ is the EWC posterior computed at the end of stage $t-1$, whereas $p(\boldsymbol{\theta} \mid \hat{\mathcal{R}}_{1:t-1}^*)$ is evaluated at stage $t$ on the estimated interference set. In this way, we account for interference without constructing $\mathcal{R}'_{1:t-1}$ at the data level.

We further adopt the Laplace approximation MacKay (1992) to model both $p(\boldsymbol{\theta}|\mathcal{R}_{1:t-1})$ and $p(\boldsymbol{\theta}|\hat{\mathcal{R}}_{1:t-1}^*)$ as Gaussian distributions centered at the converged parameters $\boldsymbol{\theta}_{t-1}^*$, with variances $\boldsymbol{\sigma}_a^2$ and $\boldsymbol{\sigma}_b^2$ derived from their respective Hessian matrices Kirkpatrick et al. (2017) $\boldsymbol{H}$ and $\boldsymbol{H}^*$. More details are provided in Appendix B.2.

Finally, by maximizing the posterior probability $\log p(\boldsymbol{\theta}|\mathcal{D}_{1:t})$ in equation 7 using Bayesian estimation, the loss at stage $t$ can be expressed as:

$$\mathcal{L}(\boldsymbol{\theta}) = \mathcal{L}_t^{det}(\boldsymbol{\theta}) + \frac{\lambda}{2} \sum_i \widetilde{\boldsymbol{I}}_i \left(\boldsymbol{\theta}_i - \boldsymbol{\theta}_{t-1,i}^*\right)^2, \quad (11)$$

where $\lambda$ is a hyperparameter that balances the plasticity and stability of the current model $\mathcal{M}_t$. $\boldsymbol{\theta}_{t-1,i}^*$ represents the parameter of the old region feature extractor $\mathcal{M}_{t-1}^F$. $\boldsymbol{\theta}_i$ represents the parameter of the current region feature extractor $\mathcal{M}_t^F$, which we aim to learn on the new data $\mathcal{D}_t$. The importance score $\widetilde{\boldsymbol{I}}_i$ for $i$-th parameter $\boldsymbol{\theta}_i$ is computed as:

$$\widetilde{\boldsymbol{I}}_i = \frac{\boldsymbol{I}_{1,i}\boldsymbol{I}_{2,i}}{(1+k)^2 \boldsymbol{I}_{2,i} + k^2 \boldsymbol{I}_{1,i}}, \quad (12)$$

with $\boldsymbol{I}_{1,i} = -\boldsymbol{H}_i$ denoting interference-carrying importance from previous tasks, capturing the sensitivity of $\boldsymbol{\theta}_i$ to old knowledge and encouraging its preservation. In contrast, $\boldsymbol{I}_{2,i} = -\boldsymbol{H}_i^*$ denotes parameter importance for interference at stage $t$, reflecting how strongly $\boldsymbol{\theta}_i$ is influenced by conflicting knowledge in the interference regions. This formulation helps preserve essential past-task parameters while relaxing constraints on those affected by interference. More details are in Appendix B.2.

## 3.4 PROTOTYPE-BASED KNOWLEDGE CALIBRATION

Although parameter regularization helps limit changes in the region feature extractor $\mathcal{M}_t^F$, it is not enough to prevent catastrophic forgetting, especially in the classification head $\boldsymbol{f}_t^{ch}(\cdot)$. This head is vulnerable to semantic drift, where the feature space shift causes old class prototypes to lose their original meaning. As the region feature space of the updated model $\boldsymbol{\theta}_t$ changes, previously learned prototypes drift, making it harder for them to represent historical class features accurately. This misalignment degrades the performance of the classification head $\boldsymbol{f}_t^{ch}(\cdot)$. To address this, we introduce a lightweight projection layer that compensates for semantic drift between the old feature space and the new feature space.

Specifically, after training the old model, we extract a set of region features $\mathcal{Q} = \{\boldsymbol{Q}_{11}, \boldsymbol{Q}_{12}, \dots, \boldsymbol{Q}_{nm}\}$ from the output of the ROI FC layers on the old data $\mathcal{D}_{t-1}$. Here, $n$ denotes the number of old classes, and $m$ is the number of region features sampled per class. Based on these features, we compute a set of class prototypes $C = \{C_1, C_2, \dots, C_n\}$. The feature distribution of each class is modeled as a Gaussian, where the mean $\boldsymbol{\mu}_i$ is computed by $\frac{1}{m} \sum_{j=1}^m \boldsymbol{Q}_{ij}$, and the variance $\boldsymbol{\sigma}_i^2$ is estimated from the diagonal of the covariance matrix $\boldsymbol{\Sigma}_i$.

Let $\mathbf{f}_t$ and $\mathbf{f}_{t-1}$ denote region features from the current task, extracted by the new and old feature extractors, respectively. The goal is to learn a projection that maps $\mathbf{f}_{t-1}$ into the feature space of $\mathbf{f}_t$. To this end, we optimize a projection loss $\mathcal{L}^{\text{proj}}$, where the projection function $\varphi(\cdot)$ is a learnable linear transformation parameterized by weight matrix $\mathbf{W}$ and bias vector $\mathbf{b}$:

$$\varphi(\mathbf{f}_{t-1}) = \mathbf{W}\mathbf{f}_{t-1} + \mathbf{b}, \quad (13)$$

$$\mathcal{L}^{\text{proj}} = \sum_{i \in \text{TopK}} \|\varphi(\mathbf{f}_{t-1,i}) - \mathbf{f}_{t,i}\|_2^2, \quad (14)$$

where $\varphi(\mathbf{f}_{t-1,i})$ denotes the projection of the old feature $\mathbf{f}_{t-1,i}$ using the projection layer, and TopK refers to the $K$ region pairs per class with the smallest L2 distances between old and new features. By minimizing $\mathcal{L}^{\text{proj}}$, we learn a transformation between the old and new region feature extractors.

Table 1: mAP@0.5 results on single incremental step on PASCAL VOC 2007. The best performance in each presented with **bold**. Methods marked with ∗ use exemplars.

| Method | Venue | 19-1 | | | 15-5 | | | 10-10 | | | 5-15 | | |
|---|---|---|---|---|---|---|---|---|---|---|---|---|---|
| | | 1-19 | 20 | 1-20 | 1-15 | 16-20 | 1-20 | 1-10 | 11-20 | 1-20 | 1-5 | 6-20 | 1-20 |
| Joint Training | - | 76.4 | 76.4 | 76.4 | 78.3 | 70.7 | 76.4 | 76.9 | 76.0 | 76.4 | 73.6 | 77.4 | 76.4 |
| Fine-tuning | - | 9.6 | 72.5 | 12.6 | 11.2 | 63.4 | 24.3 | 9.6 | 67.4 | 38.5 | 0.3 | 75.4 | 56.6 |
| ORE* Joseph et al. (2021a) | CVPR'21 | 69.4 | 60.1 | 68.9 | 71.8 | 58.7 | 68.5 | 60.4 | 68.8 | 64.6 | - | - | - |
| OW-DETR* Gupta et al. (2022) | CVPR'22 | 70.2 | 62.0 | 69.8 | 72.2 | 59.8 | 69.1 | 63.5 | 67.9 | 65.7 | - | - | - |
| ILOD-Meta* Joseph et al. (2021b) | TPAMI'21 | 70.9 | 57.6 | 70.2 | 71.7 | 55.9 | 67.8 | 68.4 | 64.3 | 66.3 | - | - | - |
| ABR* Liu et al. (2023) | ICCV'23 | 71.0 | 69.7 | 70.9 | 73.0 | 65.1 | 71.0 | 71.2 | 72.8 | 72.0 | 64.7 | 71.0 | 69.4 |
| GMDP-ABR* Wang et al. (2025b) | ICLR'25 | 74.8 | **70.1** | 74.6 | 75.8 | **65.5** | 73.2 | 72.1 | 73.2 | 72.7 | 67.1 | 71.9 | 70.7 |
| Faster ILOD Peng et al. (2020) | PRL'20 | 68.9 | 61.1 | 68.5 | 71.6 | 56.9 | 67.9 | 69.8 | 54.5 | 62.1 | 62.0 | 37.1 | 43.3 |
| PPAS Zhou et al. (2020) | - | 70.5 | 53.0 | 69.2 | - | - | - | 63.5 | 60.0 | 61.8 | - | - | - |
| MVC Yang et al. (2022) | PR'22 | 70.2 | 60.6 | 69.7 | 69.4 | 57.9 | 66.5 | 66.2 | 66.0 | 66.1 | - | - | - |
| PROB Zohar et al. (2023) | CVPR'23 | 73.9 | 48.5 | 72.6 | 73.5 | 60.8 | 70.1 | 66.0 | 67.2 | 66.5 | - | - | - |
| PseudoRM Yang et al. (2023) | MM'23 | 72.9 | 67.3 | 72.6 | 73.4 | 60.9 | 70.3 | 69.1 | 68.6 | 68.9 | - | - | - |
| MMA Cermelli et al. (2022) | CVPR'22 | 71.1 | 63.4 | 70.7 | 73.0 | 60.5 | 69.9 | 69.3 | 63.9 | 66.6 | 66.8 | 57.2 | 59.6 |
| Wu et al. Wu et al. (2024) | TIP'24 | 72.5 | 61.4 | 71.9 | 73.6 | 62.0 | 70.7 | 70.3 | 68.7 | 69.5 | - | - | - |
| BPF Mo et al. (2024) | ECCV'24 | 74.5 | 65.3 | 74.1 | 75.9 | 63.0 | 72.7 | 71.7 | 74.0 | 72.9 | 66.4 | 75.3 | 73.0 |
| GMDP-ILOD Wang et al. (2025b) | ICLR'25 | 74.2 | 67.9 | 73.9 | 74.6 | 63.5 | 71.8 | 71.9 | 69.7 | 70.8 | 65.2 | 60.5 | 61.7 |
| **Ours** | - | **76.0** | 63.2 | **75.4** | **76.9** | 64.2 | **73.7** | **75.9** | **75.5** | **75.7** | **69.7** | **77.5** | **75.6** |

After the projection layer is trained, we sample features $\mathbf{f}^s$ for old classes from a Gaussian distribution estimated from class prototypes. These features are then projected through $\varphi(\cdot)$ to obtain drift-compensated representations. We concatenate the projected features with current task features $\mathbf{f}^t$, and feed them into the classification head $\boldsymbol{f}_t^{ch}(\cdot)$ of the detection module $\mathcal{M}^{head}$ to produce logits $\boldsymbol{z}$. The classification head is then retrained using a cross-entropy loss $\mathcal{L}^{ce}$ with corresponding labels $\boldsymbol{y}_z$.

## 4 EXPERIMENTS

**Datasets and Evaluation Metrics.** We evaluate our method on the PASCAL VOC 2007 Everingham et al. (2010) and MS COCO 2017 Lin et al. (2014) datasets, following prior works Wang et al. (2025b); Mo et al. (2024); Feng et al. (2022); Dong et al. (2021); Zhou et al. (2020). PASCAL VOC 2007 contains 9,963 images across 20 categories, while COCO 2017 includes 118k training images and 5k validation images spanning 80 categories. For evaluation, we use the mean Average Precision at 0.5 IoU (mAP@0.5) for VOC, and the COCO-style mAP averaged over IoU thresholds from 0.5 to 0.95 for COCO. In each incremental setting (A-B), A denotes the number of classes in the initial stage, and B represents the number of new classes added at each subsequent stage. Columns with a gray background in the results tables report the average AP over all classes.

**Implementation Details.** Similar to previous works, our incremental object detector is built upon Faster R-CNN with a ResNet-50 He et al. (2016) backbone, initialized with ImageNet pre-trained weights Deng et al. (2009). In the VOC (10-10) setting, we set $\gamma = 0.5$, $\lambda = 20$, and $K = 32$, and sample two features per class in the PKC module. VOC experiments are conducted on a single NVIDIA GeForce RTX 4090 GPU with a batch size of 16, using PyTorch and the SGD optimizer.

### 4.1 COMPARISON WITH THE STATE-OF-THE-ART

**PASCAL VOC 2007.** We evaluate our method on the PASCAL VOC 2007 dataset using both two-stage and multi-stage incremental task settings. In the two-stage setting, tasks are divided into 19-1, 15-5, 10-10, and 5-15, where the number of new classes added at each stage increases by 1, 5, 10, and 15 classes, respectively. In the multi-stage setting, tasks are divided into 10-5, 5-5, 10-2, and 15-1, with 5, 5, 2, and 1 classes added in each step until all 20 classes are included.

**Two-stage Incremental Setting.** In Table 1, we compare our method with existing approaches. Our method consistently outperforms previous methods. Specifically, in the 19-1, 15-5, 10-10, and 5-15 settings, our method improves the average AP (Average AP among all classes) over BPF Mo et al. (2024) by 1.3%, 1.0%, 2.8%, and 2.6%, respectively, outperforms the rehearsal-based method GMDP-ABR Wang et al. (2025b) by 0.8%, 0.5%, 3.0%, and 4.9% respectively, and further surpasses GMDP-ILOD Wang et al. (2025b) by 1.5%, 1.9%, 4.9%, and 13.9% respectively. The improvements in the 1-20 show that our method achieves better stability and plasticity than most other methods.

Table 2: mAP@0.5 results on multiple incremental steps on PASCAL VOC 2007. The best performance in each presented with **bold**. Methods marked with ∗ use exemplars.

| Method | Venue | 10-5 | | | 5-5 | | | 10-2 | | | 15-1 | | |
|---|---|---|---|---|---|---|---|---|---|---|---|---|---|
| | | 1-10 | 11-20 | 1-20 | 1-5 | 6-20 | 1-20 | 1-10 | 11-20 | 1-20 | 1-15 | 16-20 | 1-20 |
| Joint Training | - | 76.9 | 76.0 | 76.4 | 73.6 | 77.4 | 76.4 | 76.9 | 76.0 | 76.4 | 78.3 | 70.7 | 76.4 |
| Fine-tuning | - | 5.3 | 30.6 | 18.0 | 0.5 | 18.3 | 13.8 | 3.8 | 13.6 | 8.7 | 0.0 | 10.5 | 5.3 |
| ABR* Liu et al. (2023) | ICCV'23 | 68.7 | 67.1 | 67.9 | 64.7 | 56.4 | 58.4 | 67.0 | 58.1 | 62.6 | 68.7 | 56.7 | 65.7 |
| GMDP-ABR* Wang et al. (2025b) | ICLR'25 | 69.9 | 67.8 | 68.9 | 66.3 | 59.3 | 61.1 | 67.6 | 58.9 | 63.3 | 69.5 | 58.9 | 66.9 |
| Faster ILOD Peng et al. (2020) | PRL'20 | 68.3 | 57.6 | 63.1 | 55.7 | 16.0 | 25.9 | 64.2 | 48.6 | 56.4 | 66.9 | 44.5 | 61.3 |
| MMA Cermelli et al. (2022) | CVPR'22 | 66.7 | 61.8 | 64.2 | 62.3 | 31.2 | 38.9 | 65.0 | 53.1 | 59.1 | 68.3 | 54.3 | 64.1 |
| BPF Mo et al. (2024) | ECCV'24 | 69.1 | 68.2 | 68.7 | 60.6 | 63.1 | 62.5 | 68.7 | 56.3 | 62.5 | 71.5 | 53.1 | 66.9 |
| GMDP-ILOD Wang et al. (2025b) | ICLR'25 | 68.1 | 62.3 | 65.2 | 61.1 | 35.8 | 42.1 | 64.2 | 53.3 | 58.8 | 66.8 | 50.4 | 62.7 |
| **Ours** | - | **72.6** | **70.3** | **71.5** | 63.8 | **66.0** | **65.4** | **69.7** | 58.4 | **64.0** | **75.8** | 42.2 | **67.4** |

Table 3: mAP results on MS COCO 2017. Methods marked with ∗ use exemplars.

| Method | Venue | 40-40 | | | 70-10 | | |
|---|---|---|---|---|---|---|---|
| | | AP | AP50 | AP75 | AP | AP50 | AP75 |
| Joint Training | - | 40.2 | 61.5 | 44.2 | 40.2 | 61.5 | 44.2 |
| Fine-tuning | - | 21.6 | 33.4 | 23.1 | 5.6 | 8.6 | 6.2 |
| ILOD-Meta* Joseph et al. (2021b) | TPAMI'21 | 23.8 | 40.5 | 24.4 | - | - | - |
| ABR* Liu et al. (2023) | ICCV'23 | 34.5 | 57.8 | 35.2 | 31.1 | 52.9 | 32.7 |
| GMDP-ABR* Wang et al. (2025b) | ICLR'25 | 36.8 | 59.6 | 36.7 | 32.5 | 53.8 | 33.9 |
| Faster ILOD Peng et al. (2020) | PRL'20 | 20.6 | 40.1 | - | 21.3 | 39.9 | - |
| PseudoRM Yang et al. (2023) | MM'23 | 25.3 | 44.4 | - | - | - | - |
| MMA Cermelli et al. (2022) | CVPR'22 | 33.0 | 56.6 | 34.6 | 30.2 | 52.1 | 31.5 |
| BPF Mo et al. (2024) | ECCV'24 | 34.4 | 54.3 | 37.3 | 36.2 | 56.8 | 38.9 |
| **Ours** | - | 35.9 | 55.8 | 38.8 | 37.1 | 57.6 | 40.6 |

Table 4: Ablation study of various values of hyper-parameters.

| Hyper-parameter | Value | VOC(10-10) | | |
|---|---|---|---|---|
| | | 1-10 | 11-20 | 1-20 |
| $\lambda$ | 1 | 73.4 | 75.2 | 74.3 |
| | 10 | 74.6 | 74.8 | 74.7 |
| | 20 | 75.0 | 75.2 | 75.1 |
| | 50 | 75.4 | 73.0 | 74.2 |
| | 100 | 75.3 | 72.4 | 73.9 |
| $K$ | 2 | 75.8 | 75.5 | 75.7 |
| | 8 | 75.8 | 75.5 | 75.6 |
| | 32 | 75.9 | 75.5 | 75.7 |
| | 128 | 75.7 | 75.5 | 75.6 |
| | 512 | 75.7 | 75.4 | 75.6 |

**Multi-stage Incremental Setting.** To further demonstrate the effectiveness of our IOD method, we also show the performance in multi-step incremental learning in Table 2. The catastrophic forgetting problem becomes more critical in longer incremental settings. Our method improves over GMDP-ABR Wang et al. (2025b) by 2.6%, 4.3%, 0.7%, and 0.5%, and outperforms BPF Mo et al. (2024) by 2.8%, 2.9%, 1.5%, and 0.5% in the 10-5, 5-5, 10-2, and 15-1 settings, respectively. It further surpasses GMDP-ILOD Wang et al. (2025b) by 6.3%, 23.3%, 5.2%, and 4.7% in the same settings. In the 15-1 setting, the dominance of old task categories leads our regularization-based method to assign higher importance to old task parameters, limiting the model's plasticity and hindering adaptation to new categories.

**MS COCO 2017.** On the challenging COCO 2017 dataset, we conduct experiments under the 40-40 and 70-10 settings, where 40 and 10 new classes are added, respectively, following the setup in Mo et al. (2024). As shown in Table 3, our method consistently outperforms the recent state-of-the-art non-rehearsal method BPF Mo et al. (2024). Under the 40-40 and 70-10 settings, our method achieves a performance gain of 1.5% and 0.9% on average AP, respectively, compared to BPF. The results demonstrate the effectiveness of our proposed method in selectively suppressing parameter-level task interference, which becomes particularly crucial as the number of classes increases.

## 4.2 ANALYSIS AND ABLATION STUDY

**Ablation Study on each component.** We evaluate the individual contributions of the IKI-EWC and PKC modules through ablation studies under the VOC 10-10, 10-5, and 5-5 settings, as shown in Table 5. Using pseudo-labeling as the baseline, IKI-EWC recalculates parameter importance in the context of incremental object detection. Specifically, pseudo labels are generated by the previous model $\mathcal{M}_{t-1}$ on current task data $\mathcal{D}_t$ to approximate annotations for old classes. IKI-EWC mitigates interference between old and new knowledge during regularization, enhancing model plasticity while effectively reducing forgetting. Compared to the baseline (rows 1 and 2), it improves performance by 1.3%, 4.5%, and 4.0% on the 10-10, 10-5, and 5-5 settings, respectively. To further reduce forgetting in the classification head, the PKC module retrains it using semantically calibrated prototypes after the final task. Compared to models without PKC (rows 1 and 3), it brings additional gains of 0.5%,

Table 5: Ablation study on each component.

| Idx | IKI-EWC | PKC | VOC(10-10) | | | VOC(10-5) | | | | VOC(5-5) | | | | |
|---|---|---|---|---|---|---|---|---|---|---|---|---|---|---|
| | | | 1-10 | 11-20 | 1-20 | 1-10 | 11-15 | 16-20 | 1-20 | 1-5 | 6-10 | 11-15 | 16-20 | 1-20 |
| 1 | – | – | 73.0 | 74.6 | 73.8 | 64.6 | 74.2 | 62.2 | 66.4 | 47.2 | 64.0 | 76.3 | 56.4 | 61.0 |
| 2 | ✓ | – | 75.0 | 75.2 | 75.1 | 72.4 | 76.4 | 62.5 | 70.9 | 62.9 | 66.6 | 75.0 | 55.6 | 65.0 |
| 3 | – | ✓ | 73.8 | 74.8 | 74.3 | 66.2 | 75.6 | 63.5 | 67.9 | 49.4 | 63.2 | 75.8 | 57.1 | 61.4 |
| 4 | ✓ | ✓ | 75.9 | 75.5 | 75.7 | 72.6 | 77.8 | 62.8 | 71.5 | 63.8 | 66.6 | 74.7 | 56.7 | 65.4 |

Table 6: Ablation study on the effect of IKI-EWC module.

| Method | VOC(10-10) | | | VOC(10-5) | | | | VOC(5-5) | | | | |
|---|---|---|---|---|---|---|---|---|---|---|---|---|
| | 1-10 | 11-20 | 1-20 | 1-10 | 11-15 | 16-20 | 1-20 | 1-5 | 6-10 | 11-15 | 16-20 | 1-20 |
| baseline | 73.0 | 74.6 | 73.8 | 64.6 | 74.2 | 62.2 | 66.4 | 47.2 | 64.0 | 76.3 | 56.4 | 61.0 |
| L2++ | 75.0 | 42.5 | 58.8 | 63.1 | 60.6 | 34.5 | 55.3 | 51.8 | 47.2 | 43.6 | 23.2 | 44.4 |
| EWC++ | 75.0 | 73.9 | 74.5 | 70.7 | 74.9 | 61.4 | 69.4 | 56.0 | 68.0 | 76.9 | 53.6 | 63.6 |
| IKI-EWC | 75.0 | 75.2 | 75.1 | 72.4 | 76.4 | 62.5 | 70.9 | 62.9 | 66.6 | 75.0 | 55.6 | 65.0 |

1.5%, and 0.4% on the three respective splits. When combined, IKI-EWC and PKC form the full IIKC framework, which consistently achieves the best performance across all settings.

**Effectiveness of the IKI-EWC Module.** We compare our parameter regularization method with EWC and L2 methods on the VOC dataset under the 10-10, 10-5, and 5-5 settings. The L2 method imposes uniform penalties on all parameters, without regard to their importance for previous tasks. Since pseudo-labeling is commonly used in prior work (e.g., IncDet Liu et al. (2020a)), excluding pseudo-labels from EWC and L2 would lead to a significant performance drop and an unfair comparison. To address this, we introduce pseudo-labels into both EWC and L2, denoted as EWC++ and L2++, respectively. As shown in Table 6, our method consistently outperforms EWC++ by 0.6%, 1.5%, and 1.4%, and surpasses L2++ by 16.3%, 15.6%, and 20.6% across the three settings, respectively. These consistent performance gains highlight the effectiveness of our regularization strategy and its superiority over traditional EWC methods, even when enhanced with pseudo-labels.

**Effectiveness of the PKC Module.** To isolate the effect of semantic drift correction, we compare the full PKC module with a variant excluding only the projection component, denoted as PKC-NPC (Table 7). The baseline refers to a model without prototype retraining and semantic alignment. Retraining with aligned prototypes yields improvements of 0.4%, 0.4%, and 1.1% on the 10-10, 10-5, and 5-5 settings, respectively, highlighting the effectiveness of correcting semantic drift in preserving performance across incremental tasks.

**Analysis Of Hyper-parameters $\lambda$ and $K$.** We analyze the impact of two key hyper-parameters: $\lambda$ and $K$. The regularization strength $\lambda$ (equation 11) balances model plasticity and stability: a smaller $\lambda$ favors plasticity but increases forgetting, while a larger $\lambda$ enhances stability at the cost of learning new tasks. As shown in Table 4, setting $\lambda = 20$ achieves the best trade-off in the 10-10 setting. The hyper-parameter $K$ (equation 14) controls the number of top-matched region feature pairs selected from the current data $\mathcal{D}_t$, based on similarity between features from the old model $\mathcal{M}_{t-1}$ and the new model $\mathcal{M}_t$. These pairs are used to train the projection layer and correct feature drift. While performance varies slightly with different $K$ values, the model is generally robust, with the best results observed at $K = 32$. Additional analyses are given in Appendix C and Appendix D.

## 5 CONCLUSION

This paper presents a novel framework for Incremental Object Detection (IOD) that combines Interference-Knowledge Isolated Elastic Weight Consolidation (IKI-EWC) with a Knowledge Calibration (PKC) strategy. Theoretically, it bridges the gap between traditional class-incremental learning and the unique challenges of IOD by building on the weight consolidation paradigm. This approach leverages new task data to identify and remove conflicting knowledge, so as to mitigate catastrophic forgetting through importance-based parameter regularization. In addition, we further propose the Prototype-based Knowledge Calibration module for region feature compensation to reduce the severity of catastrophic forgetting problem in the classification head of the object detector. Extensive experiments demonstrate that our approach effectively reduces catastrophic forgetting while maintaining high adaptability, making it a significant contribution to Incremental Object Detection.

Table 7: Ablation study on the effect of PKC module.

| Method | VOC(10-10) | | | VOC(10-5) | | | | VOC(5-5) | | | | |
|---|---|---|---|---|---|---|---|---|---|---|---|---|
| | 1-10 | 11-20 | 1-20 | 1-10 | 11-15 | 16-20 | 1-20 | 1-5 | 6-10 | 11-15 | 16-20 | 1-20 |
| baseline | 75.0 | 75.2 | 75.1 | 72.4 | 76.4 | 62.5 | 70.9 | 62.9 | 66.6 | 75.0 | 55.6 | 65.0 |
| PKC-NPC | 75.6 | 75.0 | 75.3 | 72.2 | 77.6 | 62.3 | 71.1 | 61.1 | 65.1 | 76.8 | 54.1 | 64.3 |
| PKC | 75.9 | 75.5 | 75.7 | 72.6 | 77.8 | 62.8 | 71.5 | 63.8 | 66.6 | 74.7 | 56.7 | 65.4 |

In the future, this unified, architecture-agnostic framework could be extended to detectors such as DETR, enabling broader applicability in incremental object detection.

# 6 ACKNOWLEDGMENTS

This work was supported in part by the National Key R&D Program of China under Grant No.2023YFA1008600, in part by the National Natural Science Foundation of China under Grants 62576262, U22A2096, in part by the Key Research and Development Program of Shaanxi Province under grant 2024GX-YBXM135, 2024SF-YBXM-647, in part by the Fundamental Research Funds for the Central Universities under Grant QTZX25083, QTZX23042.

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

# A  Training Pipeline for IKI-EWC Module

---

**Algorithm 1** Training Procedure of the Proposed IKI-EWC Module

---

**Input:** Task sequence $\{\mathcal{D}_1, \ldots, \mathcal{D}_T\}$ with corresponding labels $\{\mathcal{Y}_1, \ldots, \mathcal{Y}_T\}$, threshold $\gamma$, projection TopK $K$, regularization weight $\lambda$

1: **for** $t = 1$ to $T$ **do**
2:     Redefine $\mathcal{D}_t$ as $\mathcal{R}_t$
3:     **if** $t = 1$ **then**
4:         Train detection model $\mathcal{M}_t$ with general detection loss: $\mathcal{L}(\boldsymbol{\theta}) = \mathcal{L}_t^{det}(\boldsymbol{\theta})$
5:     **else**
6:         **// Step 1: Pseudo-labeling for Interference Isolation**
7:         Generate pseudo labels $\hat{\mathcal{Y}}_{1:t-1}$ for $\mathcal{D}_t$ using model $\mathcal{M}_{t-1}$
8:         Simulate prior task data $\hat{\mathcal{D}}_{1:t-1}$ by masking new-class ground truths in $\mathcal{D}_t$
9:         Identify background regions $\hat{\mathcal{R}}_{1:t-1}^-$ in $\hat{\mathcal{D}}_{1:t-1}$
10:        Estimate conflicting regions $\hat{\mathcal{R}}_{1:t-1}^* = \{i \in \hat{\mathcal{R}}_{1:t-1}^- : \exists j \in \mathcal{G}_t, \text{IoU}(i,j) \geq \gamma\}$
11:        Define $\hat{\mathcal{R}}_{1:t-1}' = \mathcal{R}_{1:t-1} \setminus \hat{\mathcal{R}}_{1:t-1}^*$
12:        **// Step 2: Compute Regularization Importance (IKI-EWC)**
13:        **for** each parameter $\theta_i$ of $\boldsymbol{\theta}_{t-1}$ **do**
14:            Compute frequency ratio $k = \frac{p(\hat{\mathcal{R}}_{1:t-1}^*)}{p(\hat{\mathcal{R}}_{1:t-1}')} = \frac{num(\hat{\mathcal{R}}_{1:t-1}^*)}{num(\hat{\mathcal{R}}_{1:t-1}')}$
15:            Compute importance $\boldsymbol{I}_{2,i}$ from conflicting knowledge $\hat{\mathcal{R}}_{1:t-1}^*$:
16:            $$\boldsymbol{I}_{2,i} = \mathbb{E}_{(x,y) \sim \hat{\mathcal{R}}_{1:t-1}^*} \left[ \left( \frac{\partial}{\partial \theta_i} \log p(y|x, \boldsymbol{\theta}) \Big|_{\theta_i = \theta_{t-1,i}^*} \right)^2 \right]$$
17:            Compute final importance: $\widetilde{\boldsymbol{I}}_i = \frac{\boldsymbol{I}_{1,i} \cdot \boldsymbol{I}_{2,i}}{(1+k)^2 \boldsymbol{I}_{2,i} + k^2 \boldsymbol{I}_{1,i}}$
18:        **end for**
19:        **// Step 3: Train Detection Model with IKI-EWC Loss**
20:        Optimize: $\mathcal{L}(\boldsymbol{\theta}) = \mathcal{L}_t^{det}(\boldsymbol{\theta}) + \frac{\lambda}{2} \sum_i \widetilde{\boldsymbol{I}}_i (\theta_i - \theta_{t-1,i}^*)^2$
21:     **end if**
22:     Get parameters $\boldsymbol{\theta}_t^*$ of model $\mathcal{M}_t$
23:     **// Step 4: Update the old importance**
24:     **for** each parameter $\theta_i$ of $\boldsymbol{\theta}_t^*$ **do**
25:         Compute Regularization Importance: $\boldsymbol{I}_i = \mathbb{E}_{(x,y) \sim \mathcal{R}_t} \left[ \left( \frac{\partial}{\partial \theta_i} \log p(y|x, \boldsymbol{\theta}) \Big|_{\theta_i = \theta_{t,i}^*} \right)^2 \right].$
26:         Update $\boldsymbol{I}_{1,i} = \boldsymbol{I}_i + \widetilde{\boldsymbol{I}}_i$
27:     **end for**
28: **end for**

---

Algorithm 1 summarizes the full training pipeline of our proposed IKI-EWC method across incremental tasks. For the first task, the model is trained using standard detection loss without any regularization, as there is no prior knowledge to preserve. Starting from the second task, we introduce a three-step process to mitigate knowledge interference: (1) *Pseudo-labeling for interference isolation*, where the old model generates labels to estimate conflict regions based on IoU with new-task ground truths; (2) *Importance estimation*, where we compute parameter-wise regularization strengths that balance old knowledge retention ($\boldsymbol{I}_{1,i}$) and suppression of conflicting gradients ($\boldsymbol{I}_{2,i}$) using the frequency-based ratio $k$; and (3) *Regularized optimization*, where the model is trained using a detection loss plus an IKI-EWC regularization term weighted by the fused importance $\widetilde{\boldsymbol{I}}_i$. After training, we update the stored importance values for use in future incremental steps. This approach allows the model to dynamically isolate interference and adaptively regulate parameter updates based on task conflict, ultimately reducing catastrophic forgetting.

## B  THEORETICAL ANALYSIS

### B.1  REGION-LEVEL POSTERIOR FACTORIZATION FOR IOD

Elastic Weight Consolidation (EWC) Kirkpatrick et al. (2017) adopts a probabilistic view of neural networks and aims to learn a model $\mathcal{M}_t$ that performs well on data from all tasks $\{\mathcal{D}_{1:t-1}, \mathcal{D}_t\}$, where $\mathcal{D}_t$ is the current-task data and $\mathcal{D}_{1:t-1}$ comprises previous tasks that are no longer accessible at stage $t$. From a Bayesian perspective, optimizing $\mathcal{M}_t$ is equivalent to maximizing the posterior $p(\boldsymbol{\theta} \mid \mathcal{D}_{1:t})$.

In incremental object detection (IOD), as shown in Figure 3, the current data $\mathcal{D}_t$ contain both future and previously seen objects yet lack complete annotations for all classes. The same limitation applies to earlier data $\mathcal{D}_{1:t-1}$, where future-class instances are unlabeled and thus treated as background. Consequently, the standard EWC factorization

$$p(\boldsymbol{\theta} \mid \mathcal{D}_{1:t}) \propto p(\mathcal{D}_t \mid \boldsymbol{\theta}) \, p(\boldsymbol{\theta} \mid \mathcal{D}_{1:t-1}) \tag{15}$$

does not hold because $\mathcal{D}_{1:t-1}$ and $\mathcal{D}_t$ are no longer (approximately) conditionally independent. Instead, we use the general form of Bayes' theorem:

$$\begin{aligned} p(\boldsymbol{\theta} \mid \mathcal{D}_{1:t}) &= \frac{p(\boldsymbol{\theta}, \mathcal{D}_{1:t})}{p(\mathcal{D}_{1:t})} \\ &= \frac{p(\boldsymbol{\theta}, \mathcal{D}_{1:t-1}, \mathcal{D}_t)}{p(\mathcal{D}_{1:t-1}, \mathcal{D}_t)} \\ &= \frac{p(\mathcal{D}_t \mid \boldsymbol{\theta}, \mathcal{D}_{1:t-1}) \, p(\boldsymbol{\theta} \mid \mathcal{D}_{1:t-1}) \, p(\mathcal{D}_{1:t-1})}{p(\mathcal{D}_t \mid \mathcal{D}_{1:t-1}) \, p(\mathcal{D}_{1:t-1})} \\ &= \frac{p(\mathcal{D}_t \mid \boldsymbol{\theta}, \mathcal{D}_{1:t-1}) \, p(\boldsymbol{\theta} \mid \mathcal{D}_{1:t-1})}{p(\mathcal{D}_t \mid \mathcal{D}_{1:t-1})}. \end{aligned} \tag{16}$$

Following Sec. 3.3, we adopt a region-based view that maps image-level data to proposal-level data for fine-grained analysis:
$$\mathcal{D}_t : \mathcal{R}_t, \qquad \mathcal{D}_{1:t-1} : \mathcal{R}_{1:t-1}.$$

Under this setting, two issues arise:

1. **Stage $t$.** We have labels for the new classes but not for old ones. Hence $\mathcal{R}_t$ contains new-class foreground regions and background regions that include old-class objects. Old-class objects are then misclassified as background, breaking independence between $\mathcal{R}_t$ and $\mathcal{R}_{1:t-1}$ and causing the same type of mismatch when training the new task.

2. **Stages $1{:}t{-}1$.** We have labels for old classes but not for the new ones. Thus $\mathcal{R}_{1:t-1}$ contains old-class foreground regions and background regions that may actually include new-class objects (interference). During proposal generation, these new-class objects are learned as background, breaking independence between $\mathcal{R}_{1:t-1}$ and $\mathcal{R}_t$. Applying EWC here would incorrectly regularize new-class objects as background.

To address the first point, we use the old model $\mathcal{M}_{t-1}$, trained on $\mathcal{D}_{1:t-1}$, to generate pseudo-labels $\hat{\mathcal{Y}}_{1:t-1}$ for old-class foreground on $\mathcal{D}_t$. These pseudo-labels prevent old-class objects from being treated as background in $\mathcal{R}_t$, thereby alleviating the non-independence between $\mathcal{R}_t$ and $\mathcal{R}_{1:t-1}$ caused by missing old labels at stage $t$. Accordingly, all subsequent derivations are based on this setup; we do not consider cases where $\mathcal{R}_t$ contains elements from the past positive set $\mathcal{R}_{1:t-1}^+$.

#### B.1.1  FOR THE SECOND POINT (IDEAL ISOLATION).

If the new-class labels $\mathcal{G}_t$ were available for past data, we would isolate the interfering regions by removing
$$\mathcal{R}_{1:t-1}^* = \left\{ r \in \mathcal{R}_{1:t-1}^- : \exists \, g \in \mathcal{G}_t, \ \mathrm{IoU}(r, g) \geq \gamma \right\}$$
from the past pool, yielding the non-conflicting subset

$$\mathcal{R}_{1:t-1}' = \mathcal{R}_{1:t-1} \setminus \mathcal{R}_{1:t-1}^*. \tag{17}$$

Consequently,

$$\mathcal{D}_{1:t}: \quad \mathcal{R}_{1:t} = \mathcal{R}'_{1:t-1} \cup \mathcal{R}_t. \tag{18}$$

By isolating the interference subset $\mathcal{R}^*_{1:t-1}$ from the past proposals, we explicitly remove conflicting knowledge. If catastrophic forgetting did not occur, a model trained sequentially—first on $\mathcal{R}'_{1:t-1}$ and then on $\mathcal{R}_t$—would behave similarly to one trained jointly on $\mathcal{R}_{1:t}$. Hence, we treat $\mathcal{R}'_{1:t-1}$ and $\mathcal{R}_t$ as approximately independent subsets, in the spirit of EWC. Applying Bayes' theorem to the region-level representation $\mathcal{R}_{1:t}$ yields

$$\begin{aligned}
p(\boldsymbol{\theta} \mid \mathcal{D}_{1:t}) &= p(\boldsymbol{\theta} \mid \mathcal{R}_{1:t}) \\
&= \frac{p(\boldsymbol{\theta}, \mathcal{R}_{1:t})}{p(\mathcal{R}_{1:t})} \\
&= \frac{p(\boldsymbol{\theta}, \mathcal{R}'_{1:t-1}, \mathcal{R}_t)}{p(\mathcal{R}'_{1:t-1}, \mathcal{R}_t)} \\
&= \frac{p(\mathcal{R}_t \mid \boldsymbol{\theta}, \mathcal{R}'_{1:t-1}) \, p(\boldsymbol{\theta} \mid \mathcal{R}'_{1:t-1}) \, p(\mathcal{R}'_{1:t-1})}{p(\mathcal{R}_t \mid \mathcal{R}'_{1:t-1}) \, p(\mathcal{R}'_{1:t-1})} \\
&= \frac{p(\mathcal{R}_t \mid \boldsymbol{\theta}, \mathcal{R}'_{1:t-1}) \, p(\boldsymbol{\theta} \mid \mathcal{R}'_{1:t-1})}{p(\mathcal{R}_t \mid \mathcal{R}'_{1:t-1})}.
\end{aligned} \tag{19}$$

Assuming approximate independence between $\mathcal{R}'_{1:t-1}$ and $\mathcal{R}_t$, i.e.,

$$p(\mathcal{R}_t \mid \boldsymbol{\theta}, \mathcal{R}'_{1:t-1}) = p(\mathcal{R}_t \mid \boldsymbol{\theta}), \qquad p(\mathcal{R}_t \mid \mathcal{R}'_{1:t-1}) = p(\mathcal{R}_t), \tag{20}$$

we obtain

$$p(\boldsymbol{\theta} \mid \mathcal{D}_{1:t}) = \frac{p(\mathcal{R}_t \mid \boldsymbol{\theta}) \, p(\boldsymbol{\theta} \mid \mathcal{R}'_{1:t-1})}{p(\mathcal{R}_t)} \propto p(\mathcal{R}_t \mid \boldsymbol{\theta}) \, p(\boldsymbol{\theta} \mid \mathcal{R}'_{1:t-1}), \tag{21}$$

which is the region-level analogue of the original EWC posterior and underpins our framework.

Finally, using the decomposition in equation 17,

$$\mathcal{D}_{1:t-1}: \quad \mathcal{R}_{1:t-1} = \mathcal{R}'_{1:t-1} \cup \mathcal{R}^*_{1:t-1}, \qquad \mathcal{R}'_{1:t-1} \cap \mathcal{R}^*_{1:t-1} = \emptyset, \tag{22}$$

we view $\mathcal{R}_{1:t-1}$ as a mixture of the non-conflicting subset $\mathcal{R}'_{1:t-1}$ and the interference subset $\mathcal{R}^*_{1:t-1}$. Accordingly, the posterior over $\boldsymbol{\theta}$ on $\mathcal{R}_{1:t-1}$ can be written as a weighted average:

$$p(\boldsymbol{\theta} \mid \mathcal{R}_{1:t-1}) = \frac{p(\mathcal{R}'_{1:t-1})}{p(\mathcal{R}_{1:t-1})} p(\boldsymbol{\theta} \mid \mathcal{R}'_{1:t-1}) + \frac{p(\mathcal{R}^*_{1:t-1})}{p(\mathcal{R}_{1:t-1})} p(\boldsymbol{\theta} \mid \mathcal{R}^*_{1:t-1}). \tag{23}$$

Since $p(\mathcal{R}_{1:t-1}) = p(\mathcal{R}'_{1:t-1}) + p(\mathcal{R}^*_{1:t-1})$, rearranging gives

$$\begin{aligned}
p(\boldsymbol{\theta} \mid \mathcal{R}'_{1:t-1}) &= \frac{p(\mathcal{R}_{1:t-1})}{p(\mathcal{R}'_{1:t-1})} p(\boldsymbol{\theta} \mid \mathcal{R}_{1:t-1}) - \frac{p(\mathcal{R}^*_{1:t-1})}{p(\mathcal{R}'_{1:t-1})} p(\boldsymbol{\theta} \mid \mathcal{R}^*_{1:t-1}) \\
&= \left( 1 + \frac{p(\mathcal{R}^*_{1:t-1})}{p(\mathcal{R}'_{1:t-1})} \right) p(\boldsymbol{\theta} \mid \mathcal{R}_{1:t-1}) - \frac{p(\mathcal{R}^*_{1:t-1})}{p(\mathcal{R}'_{1:t-1})} p(\boldsymbol{\theta} \mid \mathcal{R}^*_{1:t-1}).
\end{aligned} \tag{24}$$

### B.1.2 FOR THE SECOND POINT (IN PRACTICE).

During stages $1{:}t{-}1$, $\mathcal{G}_t$ is unavailable, so $\mathcal{R}^*_{1:t-1}$ (and thus $\mathcal{R}'_{1:t-1}$) cannot be formed at the data level. We therefore estimate the interfering subset at stage $t$ by running $\mathcal{M}_{t-1}$ on $\mathcal{D}_t$ to obtain old-class pseudo labels $\hat{\mathcal{Y}}_{1:t-1}$, constructing a simulated past set $\hat{\mathcal{D}}_{1:t-1}$ (images from $\mathcal{D}_t$ with $\hat{\mathcal{Y}}_{1:t-1}$), and passing it through $\mathcal{M}_{t-1}$ to get proposals $\hat{\mathcal{R}}_{1:t-1}$ with background subset $\hat{\mathcal{R}}^-_{1:t-1}$. We then define

$$\hat{\mathcal{R}}^*_{1:t-1} = \left\{ r \in \hat{\mathcal{R}}^-_{1:t-1} : \exists g \in \mathcal{G}_t, \ \mathrm{IoU}(r, g) \geq \gamma \right\}, \tag{25}$$

and take the surrogate non-conflicting set as

$$\hat{\mathcal{R}}'_{1:t-1} = \mathcal{R}_{1:t-1} \setminus \hat{\mathcal{R}}^*_{1:t-1}. \tag{26}$$

Let

$$k = \frac{|\hat{\mathcal{R}}^*_{1:t-1}|}{|\hat{\mathcal{R}}'_{1:t-1}|}. \tag{27}$$

Viewing the past proposals as a mixture of non-conflicting and interfering parts leads to the surrogate reweighting:

$$p(\boldsymbol{\theta} \mid \hat{\mathcal{R}}'_{1:t-1}) = (1+k)\, p(\boldsymbol{\theta} \mid \mathcal{R}_{1:t-1}) - k\, p(\boldsymbol{\theta} \mid \hat{\mathcal{R}}^*_{1:t-1}). \tag{28}$$

Here $p(\boldsymbol{\theta} \mid \mathcal{R}_{1:t-1})$ is the EWC posterior frozen at the end of stage $t-1$, and $p(\boldsymbol{\theta} \mid \hat{\mathcal{R}}^*_{1:t-1})$ is computed at stage $t$ on the estimated interference set.

## B.2    PARAMETER IMPORTANCE ESTIMATION WITH ISOLATED INTERFERENCE

To approximate the posterior $p(\boldsymbol{\theta}|\hat{\mathcal{R}}'_{1:t-1})$, we begin by applying the Laplace approximation MacKay (1992), which models probability distributions as Gaussians centered at the optimal parameters. Specifically, we approximate both the prior $p(\boldsymbol{\theta}|\mathcal{R}_{1:t-1})$ and the interference posterior $p(\boldsymbol{\theta}|\hat{\mathcal{R}}^*_{1:t-1})$ as Gaussian distributions with shared mean $\boldsymbol{\theta}^*_{t-1}$, and variances derived from their respective Hessian matrices Kirkpatrick et al. (2017):

$$\boldsymbol{\sigma}^2_a = -\frac{1}{\boldsymbol{H}}, \qquad \boldsymbol{\sigma}^2_b = -\frac{1}{\boldsymbol{H}^*}, \tag{29}$$

According to equation 28, the posterior $p(\boldsymbol{\theta}|\hat{\mathcal{R}}'_{1:t-1})$ can also be approximated as a Gaussian distribution with the same mean and a combined variance:

$$\boldsymbol{\sigma}^2_* = -\frac{\boldsymbol{H}\boldsymbol{H}^*}{(1+k)^2\boldsymbol{H}^* + k^2\boldsymbol{H}}. \tag{30}$$

We define the importance scores as $\boldsymbol{I}_1 = -\boldsymbol{H}$ and $\boldsymbol{I}_2 = -\boldsymbol{H}^*$, leading to the interference-aware importance:

$$\widetilde{\boldsymbol{I}} = \frac{\boldsymbol{I}_1\boldsymbol{I}_2}{(1+k)^2\boldsymbol{I}_2 + k^2\boldsymbol{I}_1}. \tag{31}$$

The Hessians are estimated by:

$$\begin{aligned}
\boldsymbol{H}_i &\approx -\mathbb{E}_{(x,y)\sim\mathcal{D}_{1:t-1}}\left[\left(\frac{\partial}{\partial\boldsymbol{\theta}_i}\log p(y|x,\boldsymbol{\theta})\Big|_{\boldsymbol{\theta}_i=\boldsymbol{\theta}^*_{t-1,i}}\right)^2\right], \\
\boldsymbol{H}^*_i &\approx -\mathbb{E}_{(x,y)\sim\hat{\mathcal{R}}^*_{1:t-1}}\left[\left(\frac{\partial}{\partial\boldsymbol{\theta}_i}\log p(y|x,\boldsymbol{\theta})\Big|_{\boldsymbol{\theta}_i=\boldsymbol{\theta}^*_{t-1,i}}\right)^2\right],
\end{aligned} \tag{32}$$

where these Hessians are computed at different task stages. Specifically, the Hessian $\boldsymbol{H}_i$ is estimated based on data from previous tasks $\mathcal{D}_{1:t-1}$, capturing the sensitivity of model parameters to previously learned knowledge. In contrast, the Hessian $\boldsymbol{H}^*_i$ is computed at the current task stage $t$, using the interference region set $\hat{\mathcal{R}}^*_{1:t-1}$. This set includes regions from the current task data that were incorrectly labeled as background by old model $\mathcal{M}_{t-1}$, thus representing parameter sensitivity specifically to interference knowledge.

The log-posterior over $\hat{\mathcal{R}}'_{1:t-1}$ can then be expressed as:

$$\log p(\boldsymbol{\theta}|\hat{\mathcal{R}}'_{1:t-1}) = -\frac{1}{2}\sum_i \widetilde{\boldsymbol{I}}_i(\theta_i - \theta^*_{t-1,i})^2 + C, \tag{33}$$

where $C$ is a constant. Meanwhile, the log-likelihood of the current task data is represented as:

$$\log p(\mathcal{R}_t|\boldsymbol{\theta}) = -\mathcal{L}^{det}_t(\boldsymbol{\theta}). \tag{34}$$

Combining the two terms, the posterior maximization becomes:

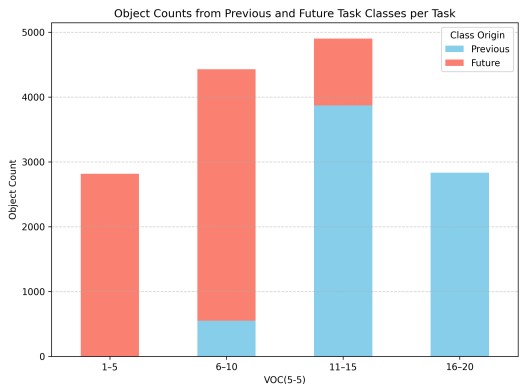

Figure 3: Object counts from previous and future task classes observed in each task under the VOC 5-5 setting. Each bar indicates the number of object instances belonging to class sets introduced in earlier tasks (blue) and those introduced in later tasks (red). This reveals inter-task interference in incremental object detection, where objects from future classes may appear before their annotations are introduced.

Table 8: Oracle analysis on VOC (15–5). mAP@0.5 is reported.

| Method | 1–15 | 16–20 | 1–20 |
|--------|------|-------|------|
| EWC | 76.8 | 61.3 | 72.9 |
| IKI-EWC | 76.7 | 63.1 | 73.3 |
| Oracle-EWC | 76.8 | 63.5 | 73.5 |

$$
\begin{aligned}
\boldsymbol{\theta} &= \arg\max_{\boldsymbol{\theta}} \left[ \log p(\mathcal{R}_t|\boldsymbol{\theta}) + \log p(\boldsymbol{\theta}|\hat{\mathcal{R}}'_{1:t-1}) \right] \\
&= \arg\min_{\boldsymbol{\theta}} \left[ \mathcal{L}_t^{det}(\boldsymbol{\theta}) + \frac{\lambda \boldsymbol{I_1} \boldsymbol{I_2}}{2(1+k)^2 \boldsymbol{I_2} + 2k^2 \boldsymbol{I_1}} (\boldsymbol{\theta} - \boldsymbol{\theta}^*_{t-1})^2 \right],
\end{aligned}
\tag{35}
$$

where $\lambda$ is a regularization coefficient. The final loss function at stage $t$ becomes:

$$
\mathcal{L}(\boldsymbol{\theta}) = \mathcal{L}_t^{det}(\boldsymbol{\theta}) + \frac{\lambda}{2} \sum_i \widetilde{\boldsymbol{I}}_i (\boldsymbol{\theta}_i - \boldsymbol{\theta}^*_{t-1,i})^2.
\tag{36}
$$

## C  ADDITIONAL ANALYSIS

### C.1  EMPIRICAL EVIDENCE OF WIDESPREAD INTERFERENCE

In IOD, the data at each incremental stage $t$ may contain objects belonging to both previous and future task classes. Objects from previous tasks appear without annotations but can be partially recovered through pseudo labeling using old detectors. In contrast, objects from future tasks remain unlabeled and are mistakenly treated as background, introducing harmful interference during training.

To quantify this phenomenon, we analyze the VOC 5-5 task split and visualize the number of object instances from both previous and future task classes encountered in each stage. As shown in Figure 3, the first two tasks observe significant amounts of future-class objects, while the latter stages receive many instances from previously seen classes. This highlights the presence of inter-task interference caused by incomplete annotations, supporting the motivation for our interference isolation strategy.

### C.2  ANALYSIS OF IDEALIZED INTERFERENCE REMOVAL

We validate interference isolation with an oracle study on VOC (15–5), where we assume perfect future labels at stage $1 : t-1$ and thus construct the ideal non-conflicting set $\mathcal{R}'_{1:t-1} = \mathcal{R}_{1:t-1} \backslash \mathcal{R}^*_{1:t-1}$

Table 9: Gradient–angle statistics (first 100 iterations, batch size 16) on VOC (10–10).

|  | $\langle g_1, g_2 \rangle$ /° | $\mathrm{Norm}(g_1)$ | $\mathrm{Norm}(g_2)$ |
|---|---|---|---|
| mean | 89.814 | 0.030 | 0.046 |
| std | 12.730 | 0.059 | 0.101 |

Table 10: Class-averaged confusion and accuracy on VOC (10–10).

| Method | Old→New (%) | New→Old (%) | Overall Mis. (%) | New Acc. (%) | Old Acc. (%) | Overall Acc. (%) |
|---|---|---|---|---|---|---|
| baseline | 11.0 | 11.2 | 11.1 | 60.5 | 57.3 | 58.9 |
| EWC | 12.2 | 10.9 | 11.5 | 61.6 | 59.0 | 60.3 |
| IKI-EWC | 11.3 | 11.4 | 11.3 | 63.2 | 59.6 | 61.4 |
| IIKC | 10.1 | 11.4 | 10.7 | 63.1 | 61.5 | 62.3 |

exactly (Oracle-EWC). We compare this to IKI-EWC, which estimates $\hat{\mathcal{R}}_{1:t-1}^*$ at stage $t$ without revisiting past data.

As shown in Table 8, IKI-EWC improves over plain EWC on the new classes (16–20) and on overall mAP, while trailing Oracle-EWC by $0.4$ (new-class) and $0.2$ (overall) mAP. Old-class accuracy (1–15) remains essentially unchanged across methods.

## C.3 Assessing Independence Between Old and New Classes

An important assumption in Sec. 3.3 is that, after isolating interference, updates from past (old-class) knowledge are approximately independent of those required by the current (new-class) task. To probe this on VOC (10–10), we measure the *angle between gradient directions* without any extra regularization.

Strictly proving *absolute* independence between two real-world data sources is infeasible: both subsets are RGB images drawn from the same visual world, so exact independence cannot be guaranteed even in classification. We therefore test for *approximate* independence by checking whether the update directions for old and new objectives are nearly orthogonal in practice.

Let $g_1$ denote the gradient of the old-task losses (classification and regression computed with old-class pseudo-labels) and $g_2$ the gradient of the new-task losses (classification and regression on new classes). Over the first 100 iterations (batch size 16), we compute the angle $\langle g_1, g_2 \rangle$ (degrees) and the norms $\|g_1\|$, $\|g_2\|$. The statistics are summarized in Table 9: the mean angle is $89.8°$ with a standard deviation of $12.7°$, and the mean norms are $\|g_1\| = 0.030$ and $\|g_2\| = 0.046$ (std 0.059 and 0.101).

Angles near $90°$ indicate that the two update directions are close to orthogonal, implying weak mutual interference. These results support our working assumption that, once interference regions are isolated, $\mathcal{R}'_{1:t-1}$ and $\mathcal{R}_t$ can be treated as approximately independent for posterior factorization.

## C.4 Confusion and Accuracy Analysis Between Old and New Classes

Using Table 10, we assess whether EWC amplifies *old–new foreground* confusion and whether PKC mitigates it. Overall, plain EWC produces only minor changes in the aggregate misclassification rate, suggesting it does not substantially worsen cross–task confusion. In contrast, adding PKC within our full system (IIKC) lowers the overall confusion between old and new foreground categories and is accompanied by a clear improvement in both new-class and old-class accuracy. In short, EWC has a limited effect on *old–new foreground* confusion, while PKC effectively reduces it and improves accuracy.

Table 11: Impact of Different $k$ Values on IKI-EWC Performance.

| Value of k | VOC(10-10) | | |
|---|---|---|---|
| | 1-10 | 11-20 | 1-20 |
| 0.0 | 75.0 | 73.9 | 74.5 |
| 0.0087 | 75.0 | 75.2 | 75.1 |
| 868183.4375 | 74.3 | 74.4 | 74.3 |

Table 12: Ablation study on the IoU threshold $\gamma$ under VOC (10–10).

| Hyper-parameter | Value | VOC (10–10) | | |
|---|---|---|---|---|
| | | 1–10 | 11–20 | 1–20 |
| $\gamma$ | 0.20 | 74.7 | 75.2 | 75.0 |
| | 0.35 | 75.1 | 74.6 | 74.9 |
| | 0.50 | 75.0 | 75.2 | 75.1 |
| | 0.65 | 75.1 | 74.8 | 75.0 |
| | 0.80 | 75.5 | 74.2 | 74.9 |

## C.5 ANALYSIS OF SCALING FACTOR $k$ AND HYPER-PARAMETER $\gamma$

### C.5.1 THE SCALING FACTOR $k$

The interference-aware regularization term in equation 35 is defined as:

$$\boldsymbol{\theta} = \arg\min_{\boldsymbol{\theta}} \left[ \mathcal{L}_t^{det}(\boldsymbol{\theta}) + \frac{\lambda \boldsymbol{I}_1 \boldsymbol{I}_2}{2(1+k)^2 \boldsymbol{I}_2 + 2k^2 \boldsymbol{I}_1} (\boldsymbol{\theta} - \boldsymbol{\theta}_{t-1}^*)^2 \right], \tag{37}$$

where the scaling factor $k = \frac{p(\hat{\mathcal{R}}_{1:t-1}^*)}{p(\hat{\mathcal{R}}_{1:t-1}')}$ reflects the relative prevalence of interference in the old data. Intuitively, a larger $k$ implies more contamination by future-class knowledge, thus requiring stronger down-weighting of corresponding parameters during regularization.

To validate the theoretical impact of $k$, we conduct experiments under the VOC 10-10 setting, comparing three configurations: (1) the computed $k$, (2) $k = 0$ (ignoring interference), and (3) an excessively large $k$ scaled by $10^8$. As illustrated in Figure 4 and Table 11, the computed $k$ yields the best performance across all class splits, while both extreme values degrade results. This confirms that $k$ captures the trade-off between stability and interference suppression, validating the effectiveness of our formulation.

### C.5.2 THE HYPER-PARAMETER $\gamma$

The IoU threshold $\gamma$ (see equation 8) determines whether a current foreground region is treated as background by the old detector. As shown in Table 12, the metrics change only slightly as $\gamma$ varies.

## D COMPUTATIONAL OVERHEAD ANALYSIS

Table 13 summarizes efficiency on VOC (10–10) with a single RTX 4090. We compare plain EWC, our two modules used separately (IKI-EWC as module 1 and PKC as module 2, a lightweight projection layer removed after training), the full IIKC system, and a strong non-replay baseline (BPF; the first row shows our re-implementation). GFLOPs are reported per iteration (forward plus backward) with inputs of size $3 \times 608 \times 928$, and "Additional GPU" denotes the extra memory beyond the stored checkpoint. The results indicate that IKI-EWC and PKC each add only modest computation and memory while maintaining overall efficiency, and the combined IIKC system achieves the highest accuracy with only a small cost increase relative to EWC. We also report runtime on COCO in Table 14.

## E LIMITATION

A limitation of IKI-EWC is its reliance on the old model for generating pseudo labels, which may lead to imprecise annotations and residual interference. This issue could potentially be addressed

Table 13: Efficiency comparison under VOC (10-10) on a single NVIDIA RTX 4090.

| Method | 1–20 mAP @ 0.5 | GFLOPs | Running time (min) | Additional GPU (MB) | #Params (M) |
|---|---|---|---|---|---|
| BPF (ECCV'24) | 72.9 | 369 | 364 | 252.07 | 33.2 |
| EWC | 74.5 | 333 | 173 | 313.78 | 41.4 |
| IKI-EWC (first module) | 75.1 | 333 | 174 | 313.78 | 41.4 |
| PKC (second module) | 74.3 | 6 | 15 | 4.01 | 1.05 |
| **IIKC (full system)** | 75.7 | 339 | 189 | 317.79 | 42.5 |

Table 14: Running time on COCO (70-10) with Two RTX 4090 GPUs.

| Method | Running time (minutes) |
|---|---|
| IKI-EWC | **2499.14** |

Figure 4: Ablation study of the scaling factor $k$ in IKI-EWC. We compare the computed optimal value with two extreme settings: a minimal value $k = 0$ and an excessively large value scaled by $10^8$. Results show that a properly estimated $k$ leads to the best performance, validating its theoretical derivation.

in the future by incorporating memory replay techniques to better preserve old task representations and integrating attention mechanisms to reduce misclassification errors between foreground and background regions.

# F  POTENTIAL IMPACT

Our method enhances the adaptability of object detection systems in dynamic environments, benefiting applications such as autonomous driving, assistive robotics, and privacy-preserving surveillance by reducing the need to store or retrain on historical data. However, as with many machine learning techniques, it may be misused in unethical surveillance or military contexts, and remains susceptible to bias, imbalance, or adversarial data if deployed without adequate safeguards.

# G  LARGE LANGUAGE MODEL USAGE STATEMENT

We relied on a large language model (LLM) only for language refinement (grammar, wording, and clarity). The LLM did not contribute to the conception of ideas, study design, data collection, analysis, or figure/table generation, and it did not write technical content. All methodological and experimental contributions, as well as the interpretation of results and final decisions, are by the authors.

