# OpenReview forum: "Interference-Isolated Elastic Weight Consolidation and Knowledge Calibration for Incremental Object Detection"
_ICLR.cc/2026/Conference — ICLR 2026 Poster_

### Official Review · Reviewer_a9JU · 2025-10-28

**Soundness:** 3
**Presentation:** 3
**Contribution:** 3
**Rating:** 6
**Confidence:** 3

**Summary:**

This paper addresses a key issue in incremental object detection: future classes are unlabeled in early tasks and are therefore learned as background, but later must be detected as foreground. This background conflict causes strong interference and catastrophic forgetting. The proposed framework consists of IKI-EWC, which aims to isolate and down-weight conflicting background knowledge during consolidation, and PKC, which aligns stored old-class feature prototypes to the current feature space and recalibrates the current classification head without keeping raw past images.

**Strengths:**

1.	The paper focuses on a real problem in incremental detection: future classes are treated as background first, then must be detected later. It proposes two modules, IKI-EWC and PKC, to keep old knowledge without storing past images.
2.	Experiments on PASCAL VOC and MS-COCO are broad, and ablations show both modules are useful.

**Weaknesses:**

1.	Compared with current state-of-the-art methods.
The paper claims state-of-the-art performance relative to prior incremental detection methods, but it does not compare against recent approaches such as RGR[1] and GMDP-ABR[2], which report equal or stronger final mAP on both multi-step PASCAL VOC and MS-COCO splits. The paper should include a direct comparison to RGR and GMDP-ABR in the main tables and clearly state in which regimes the proposed method is preferable, for example, no generator cost, lower complexity, or better stability on old classes.
2.	IKI-EWC formulation clarity.
IKI-EWC is presented as deriving a clean posterior by separating non-conflicting and conflicting regions and then using this to define a new importance term for an EWC-style penalty. However, the core assumptions behind this construction, for example, proposal-level independence, using the previous model to approximate past label structure on current data, and a Laplace or Gaussian approximation, are only implicit. These assumptions should be stated explicitly in the main text where the final loss is introduced.
3.	Memory usage.
The paper emphasizes that it does not store past images, but PKC does maintain a feature memory of sampled ROI features and Gaussian prototypes for old classes. The total storage cost of this memory, for example, the number of stored features per class, their dimensionality and total size, is not reported, and there is no quantitative comparison to exemplar replay or to generative replay methods, which also claim to avoid storing raw past data but still keep some form of replay budget.  Reporting the memory footprint would make the comparison stronger and more credible.

[1] Revisiting Generative Replay for Class Incremental Object Detection

[2] HIGH-DIMENSION PROTOTYPE IS A BETTER INCREMENTAL OBJECT DETECTION LEARNER

**Questions:**

1.	Please add current state-of-the-art methods to the PASCAL VOC and MS-COCO comparisons.
2.	Please state the explicit assumptions used in the IKI-EWC derivation in the main text.
3.	Please report the prototype buffer size and memory usage.

---

> ### Author Response · Authors · 2025-11-19
>
> We sincerely thank the reviewer for the constructive suggestions and the positive assessment of our work. We will update our paper.
>
> **W1 & Q1:**
> We thank the reviewer for the suggestion. Our paper already compares against the recent GMDP-ABR in Tables 1–3. Below we further contrast GMDP-ABR, RGR, and our method.
>
> **Pipeline during new-task training:**
>
> | Method   | Procedure (VOC 10-10) |
> |----------|-----------|
> | GMDP-ABR | Stores old-task region crops (15.5 MB). Trains $M^t$ on new-task images together with the stored old regions. |
> | RGR      | Fine-tunes Stable Diffusion v1.5 on COCO for 50 epochs. Trains two auxiliary detectors $M^{t-1}$ and $M^{s\_t}$. Uses diffusion to generate images for old and new tasks, labels them with $M^{t-1}$ and $M^{s\_t}$, then trains $M^t$ on generated images plus new-task data. |
> | IIKC     | Trains $M^t$ on new-task data only. No generator. No extra stored regions. |
>
> **MS-COCO 70–10:**
>
> | Method   |Venue| AP   | AP$\_{50}$ | AP$\_{75}$ |
> |----------|------|------|-------------|-------------|
> | RGR (uses replayed data) | CVPR'25 | 36.6 | 56.6        | 39.6        |
> | GMDP-ABR (uses replayed data) | ICLR'25| 32.5 | 53.8        | 33.9        |
> | IIKC (Ours) | --   | 37.1 | 57.6        | 40.6   |
>
> **MS-COCO 40–40:**
>
> | Method   |Venue| AP   | AP$\_{50}$ | AP$\_{75}$ |
> |----------|------|------|-------------|-------------|
> | RGR (uses replayed data)  | CVPR'25   | 35.6 | 56.0        | 38.0        |
> | GMDP-ABR (uses replayed data) | ICLR'25 | 36.8 | 59.6        | 36.7        |
> | IIKC (Ours)    | -- | 35.9 | 55.8        | 38.8        |
>
> **PASCAL VOC:**
>
> | Method   |Venue| 10–10 (two tasks) | 5–15 (two tasks) | 10–5 (three tasks) |
> |----------|------|--------------------|------------------|--------------------|
> | RGR (uses replayed data)  | CVPR'25   | 75.8               | 75.3             | 71.8               |
> | GMDP-ABR (uses replayed data) | ICLR'25 | 72.7               | 70.7             | 68.9               |
> | GMDP-ILOD | ICLR'25 | 70.8 | 61.7 | 65.2 |
> | IIKC (Ours)  | -- | 75.7               | 75.6             | 71.5               |
>
> These results show that IIKC matches or exceeds RGR and GMDP-ABR across VOC and COCO settings while avoiding generator training and extra storage. Our method favors regimes that require lower complexity and stable old-class retention without the cost of diffusion fine-tuning or maintaining region banks.
>
> **W2 & Q2:**
> Thank you for the helpful suggestion. We agree that the core assumptions should be stated explicitly where the final loss is introduced. We will add a short paragraph that lists:
>
> 1) **Approximating past label structure on current data.** At stage $t$ we use the previous detector to estimate the interfering subset $\hat R^{\*}\_{1:t-1}$ on $D\_t$ with new classes masked and define the relative mass $k$. This enables posterior reconstruction without revisiting past raw data.
>
> 2) **Region-level factorization after removing interference.** After splitting proposals into non-conflicting $R'\_{1:t-1}$ and conflicting $R^{\*}\_{1:t-1}$ we assume $R'\_{1:t-1}$ is approximately independent of current proposals $R\_t$.
>
> 3) **Laplace approximation.** We adopt a MAP objective with a Laplace approximation around $\theta^{\*}\_{t-1}$. This yields a quadratic penalty with a fused interference-aware importance $\tilde I$ in Eq.12.
>
> **W3 & Q3:**
> We agree with the reviewer’s concern. We store only a per class prototype mean $\mu$ of dimension $d$ and a single scalar that is the mean of the trace of the covariance. The total footprint is 44.6 KB.
>
> For comparison, according to the GMDP-ABR paper, each class keeps two means of dimension $d$ and two variance terms of dimension $d\times d$, and the method additionally replays 2,000 old task region crops, which occupy 15.5 MB. As a simple reference, storing one mean of dimension $d$ and one $d\times d$ variance per class would already consume about 44.04 MB.
>
> **Memory footprint summary (VOC 10-10):**
>
> | Method      | What is stored                                                                                           | Total size |
> |-------------|-----------------------------------------------------------------------------------------------------------|------------|
> | PKC (Ours)  | per class mean $\mu$ $(d)$ and a scalar covariance trace                                              | 44.6 KB    |
> | GMDP-ABR    | two prototype means $(d)$ and two variance tensors $(d\times d)$ per class, plus 2,000 region crops   | 15.5 MB for crops, prototype statistics additional (≈ 88.08 MB) to this |
> | Reference   | one mean $(d)$ and one variance $(d\times d)$ per class                                               | 44.04 MB |

---

> > ### Comment · Reviewer_a9JU · 2025-11-27
> >
> > The author's response resolved my concerns, and I raise my score.

---

> > > ### Author Response · Authors · 2025-11-27
> > >
> > > We sincerely thank the reviewer for the positive assessment and for raising the score. We truly appreciate your time and constructive feedback.

---

### Official Review · Reviewer_Zo9q · 2025-10-29

**Soundness:** 2
**Presentation:** 3
**Contribution:** 2
**Rating:** 4
**Confidence:** 4

**Summary:**

Authors introduce IIKC, a novel framework for Incremental Object Detection (IOD) that combines interference-aware Bayesian regularization (IKI-EWC) and Prototype-based Knowledge Calibration (PKC) to tackle catastrophic forgetting and knowledge conflict during continual learning of new object classes. IIKC identifies and isolates regions in new task data that create interference, using pseudo labels from the old detector, and recalculates parameter regularization based on both retained and conflicting knowledge. The PKC module corrects semantic drift by realigning previous class features with current ones using a learnable projection, retraining the classifier on calibrated prototypes. Experiments on PASCAL VOC and MS-COCO benchmarks show IIKC consistently outperforms state-of-the-art regularization and rehearsal methods on incremental settings, with higher mAP and reduced forgetting across more challenging task splits.

**Strengths:**

1) Novel Theoretical Contribution: The paper makes a meaningful theoretical contribution by reformulating Elastic Weight Consolidation (EWC) in a Bayesian framework that explicitly accounts for interference knowledge — regions where unlabeled objects from future classes are mistakenly learned as background. This provides a principled mechanism to isolate and suppress task conflicts, addressing a long-standing limitation in incremental object detection (IOD).

2) Comprehensive Framework: By integrating two complementary components — IKI-EWC for parameter-level stability and PKC for feature-level calibration — the proposed IIKC framework tackles both catastrophic forgetting and semantic drift. This dual approach effectively bridges low-level model regularization with high-level feature alignment, demonstrating thoughtful architectural design.

**Weaknesses:**

1) Dependence on Pseudo-Labels and Sensitivity to Noise: The IKI-EWC module is heavily dependent on pseudo-labels provided by the former detector to estimate interference regions. In the incremental object detection setting, these pseudo-labels are usually noisy, particularly for classes where the previous model has low performance. Subsequently, incorrect pseudo-labels may misidentify interference regions and subsequently impact the estimation parameter importance. The manuscript does not qualify any pseudo-label accuracy or sensitivity studies to suggest feasibility under varying degrees of noise for the IKI-EWC to evaluate stability. It is difficult to assess feasibility in more realistic scenarios without assessing effect when the pseudo-labels are not accurate.

2) Computational Complexity and Scalability Concerns: The proposed framework has several computation-heavy procedures, including running the old model on all data for new tasks for pseudo-labels, computing interference ratios (𝑘), and estimating Fisher-based parameter importance for large networks such as Faster R-CNN. All operations can be both cost-prohibitive on memory, and time. The paper does not indicate training overhead, runtime comparison, or scalability analysis for the number of classes or incremental steps are increased (e.g., COCO 80 classes or LVIS 1200+ classes), which raises concerns for practicality of the method
3) Residual interference: Future-class objects remain unlabeled and are still at risk of being treated as background, which may not be fully resolved by the current method.

IIKC offers promising advances for incremental detection, balancing stability and plasticity, but future work could improve robustness by incorporating memory replay or better foreground-background attention mechanisms.

**Questions:**

1) Novelty & Conceptual Soundness: The paper introduces IKI-EWC to isolate interference knowledge in incremental object detection, but it remains somewhat unclear how this approach fundamentally differs, both theoretically and algorithmically, from prior interference-aware frameworks such as BPF (Mo et al., 2024) and GMDP (Wang et al., 2025), which also attempt to mitigate background conflicts and feature drift; could the authors provide a deeper explanation of what new insight of the reformulation and posterior correction (Eq. 10) contribute beyond existing EWC-based or distillation-based methods, and whether this formulation yields measurable theoretical guarantees or only empirical improvements?

2) Methodology & Implementation: The proposed interference isolation depends heavily on pseudo-labels generated by the previous model to identify conflicting regions; since pseudo-label quality can vary widely and introduce noise, could the authors analyze how the accuracy of these pseudo-labels impacts interference estimation, describe the computational overhead of running the old detector on all new data, and clarify how the approach scales to large datasets (e.g., COCO or LVIS) where the number of proposals and IoU computations may become prohibitively expensive?

3) Equation 2 and 3 mostly focus on Bayesian posterior? Is your framework Bayesian or leverage variational inference to build the network?

4) Overall optimization is not clear.  Equation 2 is similar to EWC. The only difference is you replaced diagonal Fisher matrix with equation 12?. And, section 3.4: Equation 14 is just knowledge distillation on features? How this norm is helping for knowledge calibration? The total loss is combination of these two loss?

---

> ### Author Response · Authors · 2025-11-19
>
> We sincerely thank the reviewer for the insightful comments and for appreciating our approach.
>
> **W1 & Q2:**
> Thank you for the thoughtful question. As noted in our limitations, our method, like prior IOD work, is affected by pseudo-label quality. Following prior pseudo-label–based studies [2][3][4], which are themselves sensitive to pseudo-label quality, we use a unified and strong pseudo-label configuration across all experiments to control this factor and enable fair comparison.
> Conceptually, **Eq. 8** applies a dual gate that combines pseudo labels with the ground truth for the current task. The ground truth anchors the mask, so even a substantial drop in pseudo-label quality does not cause severe failure.
>
> **Eq. 8:**
> $$
> \hat{\mathcal{R}}\_{1:t-1}^{*} = \lbrace{ r \in \hat{\mathcal{R}}\_{1:t-1}^{-}\ :\ \exists g \in \mathcal{G}\_t,\ \mathrm{IoU}(r,g)\ge \gamma \rbrace}.
> $$
>
> To better approximate realistic noise, we used three undertrained models as the previous detector at epochs 2, 4, and 6 on VOC 10–10. The row **base** reports the first‐task performance of that old model. The row **k** reports the estimated interference ratio under that model. The row **baseline** applies a common pseudo‐labeling strategy from IOD. The row **IKI‐EWC** reports our method under the same setting.
>
> |                | epoch 2 | epoch 4 | epoch 6 |
> |----------------|---------|---------|---------|
> | base           | 49.7    | 69.7    | 72.8    |
> | $k$              | 0.0013  | 0.0025  | 0.0021  |
> | baseline       | 55.2    | 69.2    | 71.4    |
> | IKI-EWC        | 57.1(+1.9)   | 70.8 (+1.6)   | 72.5 (+1.1)   |
>
> Earlier checkpoints yield pseudo labels with lower precision and recall than later ones. These results indicate that IKI-EWC remains stable when pseudo‐labels are noisy and consistently outperforms the baseline at matching noise levels.
>
> We also vary the pseudo-label confidence threshold when computing IKI-EWC to simulate different pseudo-label qualities. Lower thresholds increase recall but reduce precision and introduce more noisy positives. Even at a low threshold of 0.2 where pseudo labels are very noisy, the performance remains above the baseline of 73.8 mAP.
>
> **VOC 10–10:**
>
> | Confidence threshold | 1–20 (mAP) |
> |------------------|----------|
> | 0.20                 | 74.4 (+0.6)    |
> | 0.50                 | 74.7 (+0.9)   |
> | 0.75                 | 75.1  (+1.3)   |
>
> **W3:** Thank you for raising this point. Residual interference arises because future classes are unknown in IOD. As we state in the paper, accessing information about future classes is not feasible. We therefore propose IKI-EWC, which removes interference for task $t$ during the training of task $t$. For clarity, we refer to task $t$ as the new task and tasks $1{:}t{-}1$ as the old tasks. When task $t{+}1$ arrives, tasks $1{:}t$ become the old tasks and $t{+}1$ becomes the new task, and the same formulation applies. Appendix Algorithm 1 provides the end-to-end training procedure from the first task to the $T$-th task.
>
> **Q1:** Thank you for the question. BPF trains two separate teachers for the old and new tasks and distills the student by combining their probability outputs. GMDP reconstructs class prototypes as a Gaussian mixture and samples prototypes during the new task to suppress feature drift. Our method IKI-EWC rederives EWC for detection and makes interference in IOD explicit in parameter space. We reconstruct the posterior and obtain a closed-form fused importance $\tilde I$. The derivation is a MAP-based interference isolation that acts on model parameters, rather than aligning outputs or features as in BPF and GMDP. Although all three aim to address IOD, the principles and algorithms are different.
>
> **Eq 6:**
> $\mathcal{D}\_{1:t}
> = \big(\mathcal{R}\_{1:t-1}\setminus \mathcal{R}\_{1:t-1}^{*}\big)\ \cup\ \mathcal{R}\_{t}
> = \mathcal{R}\_{1:t-1}^{\prime}\ \cup\ \mathcal{R}\_{t}.$
>
> **Eq 10:**
> $p(\boldsymbol{\theta}\mid \hat{\mathcal{R}}\_{1:t-1}') = (1+k) p(\boldsymbol{\theta}\mid \mathcal{R}\_{1:t-1}) - k p(\boldsymbol{\theta}\mid \hat{\mathcal{R}}\_{1:t-1}^{*}).$
>
> Eq 10 builds on the non-conflict and conflict partition in Eq 6 and the relative mass $k$. It expresses the unavailable clean posterior using available posteriors, then applies a Laplace approximation to yield the closed form $\tilde I$. This gives a MAP objective that isolates interference when the old model can conflict with future knowledge.
>
> The formulation has clear boundary behavior. When $k=0$, there is no interference and Eq 10 reduces to standard EWC. When $k$ becomes very large, Eq 12 imposes stronger down-weighting on the parameters during regularization. Appendix C.5.1 provides the detailed discussion. Empirically, our results on the VOC protocols in the main text, including 10–10 with two tasks, 10–5 with three tasks, and 5–5 with four tasks, support the effectiveness of the approach.

---

> ### Author Response · Authors · 2025-11-19
>
> **W2 & Q2:**
> We appreciate the reviewer’s concern. In our pipeline, running the old model for pseudo labels, computing the interference ratio $k$, and estimating Fisher based parameter importance are one pass operations executed in a single offline sweep. The ratio $k$ is a simple count statistic and adds negligible compute. Relative to the training time, this preprocessing is small.
>
> In Faster R-CNN training, the number of proposals per image is capped to a fixed budget for the second stage head, commonly 512. Our interference ratio $k$ is computed on these capped proposals after NMS, so the per image workload is $O(N)$ with $N$ constant. This makes the computation linear in the number of images and essentially independent of the number of classes. The IoU computations are vectorized and share the same cost profile as a standard forward pass. We do not store proposals. We only count how many proposals satisfy the IoU criterion with the current ground truth, which keeps memory usage low. In practice, this step is a single offline sweep over the dataset and its runtime is negligible compared with end to end training.
>
> **Cost of the one pass preprocessing on a single RTX 4090 under VOC 10–10:**
>
> | Operation                                  | Time     | GPU memory | CPU memory |
> |--------------------------------------------|----------|------------|------------|
> | Pseudo labeling + $k$ computation + Fisher | 256 s | 19,108 MB  | 313.78 MB  |
>
> The training time for VOC 10–10 is **10440 s** in Appendix Table 13.
>
> **Cost of the one pass preprocessing on a single RTX 4090 under COCO 70–10:**
>
> | Operation                                  | Time      | GPU memory | CPU memory |
> |--------------------------------------------|-----------|------------|------------|
> | Pseudo labeling + $k$ computation + Fisher | 56 min 50 s | 20,902 MB  | 313.78 MB  |
>
> The training time for COCO 70–10 is **4,998 min** in Appendix Table 14.
>
> The remaining training overhead and runtime details are reported in Appendix Table 13.
>
> **Table 13:**
> | Method     | GFLOPs | Running time (min) | Additional GPU (MB) | #Params (M) |
> |--------------|--------|-------------|-----------|-------------|
> | BPF| 369    | 364  | 252.07 | 33.2 |
> | EWC |  333 | 173 | 313.78 | 41.4  |
> | IKI-EWC (first module) | 333| 174 | 313.78  | 41.4  |
> | PKC (second module)   | 6  | 15 | 4.01  | 1.05 |
> | IIKC (full system)   | 339 | 189 | 317.79 | 42.5  |
>
> **Scalability evidence:**
>
> We follow standard incremental protocols on VOC with 10–5 over three tasks, 5–5 over four tasks, 10–2 over six tasks, and 15–1 over six tasks as in Table 2. We also evaluate on COCO with 80 classes under 40–40 and 70–10 as in Table 3.
>
> |  | 10-5 (3 tasks) | 5-5 (4 tasks) |
> |---------------------|--------|-------------|
> | baseline            | 66.4   | 61.0       |
> | IKI-EWC             | 70.9   | 65.0       |
>
> In addition, we run a three task COCO 40–20 setting. Results are below.
>
> |   | $AP$ | $AP\_{50}$ | $AP\_{75}$ |
> |---------------------|--------|-------------|-------------|
> | baseline            | 33.0   | 52.0        | 36.3        |
> | IKI-EWC             | 33.6   | 52.4        | 36.8        |
>
> These results indicate that the preprocessing step is lightweight and that the method scales across more tasks and larger class counts.
>
> **Q3:** Thank you for raising this point. Eq 2 and 3 revisit the posterior factorization that underpins EWC and show that conditional independence can break in IOD due to interference, which motivates our region-level reconstruction. These equations serve as the theoretical starting point and do not claim the use of variational inference. Our method is Bayesian inspired maximum a posteriori regularization[1] rather than a full Bayesian network. We do not learn a distribution over weights and we do not use variational inference. We optimize a MAP objective with a Laplace approximation, which yields an EWC based quadratic penalty whose coefficients are interference aware.

---

> > ### Author Response · Authors · 2025-11-19
> >
> > **Q4:**
> > Thank you for the detailed question.
> > (**Eq 3:**)
> > $p(\boldsymbol{\theta}|\mathcal{D}\_{1:t}) \propto p(\mathcal{D}\_t|\boldsymbol{\theta})  p(\boldsymbol{\theta}|\mathcal{D}\_{1:t-1})$
> > presents the EWC objective under a MAP formulation. It views the parameter posterior from a Bayesian perspective and factors it into task related components. In incremental object detection, however, past data may contain unlabeled future classes and current data may contain unlabeled past classes. The condition behind Eq. 3 does not hold. We therefore fall back to Eq. 2.
> >
> > **Eq 2:**
> > $p(\boldsymbol{\theta}\mid \mathcal{D}\_{1:t}) \propto p(\mathcal{D}\_t \mid \boldsymbol{\theta}, \mathcal{D}\_{1:t-1}) p(\boldsymbol{\theta}\mid \mathcal{D}\_{1:t-1}) .$
> >
> > Eq 2 is not directly usable for further factorization in this setting. So we reconstruct it for IOD. Our final result is Eq. 12. It is not a simple or ad hoc substitution. It is consistent with the MAP starting point. It follows from the steps below:
> >
> > For IKI-EWC, we map images to the region proposal sets used by detection. In the ideal case, we obtain the region level EWC factorization with the clean set $\mathcal{R}\_{1:t-1}'$:
> >
> > **Eq 7:**
> > $p(\boldsymbol{\theta}\mid \mathcal{D}\_{1:t}) \propto p(\mathcal{R}\_{t}\mid \boldsymbol{\theta})p(\boldsymbol{\theta}\mid \mathcal{R}\_{1:t-1}^{\prime}).$
> >
> > Because interference exists, $\mathcal{R}\_{1:t-1}\neq \mathcal{R}\_{1:t-1}'$. Let the interfering subset be $\mathcal{R}\_{1:t-1}^{*}$. Then
> >
> > **Eq 24:**
> > $p(\boldsymbol{\theta}\mid \mathcal{R}\_{1:t-1}^{\prime})=(1 + \frac{p(\mathcal{R}\_{1:t-1}^{\*})}{p(\mathcal{R}\_{1:t-1}^{\prime})}) p(\boldsymbol{\theta} \mid \mathcal{R}\_{1:t-1}) - \frac{p(\mathcal{R}\_{1:t-1}^{\*})}{p(\mathcal{R}\_{1:t-1}^{\prime})} p(\boldsymbol{\theta}\mid \mathcal{R}\_{1:t-1}^{\*}) .$
> >
> > At stage $t$, we estimate the $p(\boldsymbol{\theta}\mid \hat{\mathcal{R}}\_{1:t-1}^{\*})$ using only current data and pseudo labels from the old model. And we do not revisit past raw data.
> > Next, we apply a Laplace approximation to $p(\boldsymbol{\theta}\mid \mathcal{R}\_{1:t-1})$ and $p(\boldsymbol{\theta}\mid \hat{\mathcal{R}}\_{1:t-1}^{*})$. This yields the combined curvature and the fused importance
> >
> > **Eq 12:**
> > $\tilde I = \frac{I\_1\,I\_2}{(1+k)^2 I\_2 + k^2 I\_1} , k=\frac{p(\hat{\mathcal{R}}\_{1:t-1}^{\*})}{p(\mathcal{R}\_{1:t-1}^{\prime})}$
> >
> > Here $I\_1=-H$ comes from historical tasks and $I\_2=-H^{*}$ comes from the estimated interfering subset. Both are estimated with diagonal Fisher. The full derivation is provided in Appendix B.
> >
> > **Eq 14:**
> > $\mathcal{L}^\text{proj} = \sum\_{i \in {\mathrm{TopK}}} \left\| \varphi(\mathbf{f}\_{t-1,i}) - \mathbf{f}\_{t,i} \right\|\_2^2.$
> >
> > Eq 14 is a knowledge calibration objective rather than generic feature distillation. During training on the new task the feature extractor drifts. Even if old task prototypes are stored they can become mismatched because region features shift under the updated backbone or head. We therefore learn a projection layer that compensates the drift between same class region features produced by the old and new models for the same image. Eq 14 trains this projection so that prototypes are semantically adjusted by the projection and the calibrated prototypes can be used to retrain the classification head.
> >
> > The total loss combines these two terms with $L^{ce}$, which is the cross entropy loss used to retrain the classification head.
> >
> > [1] Kirkpatrick J, Pascanu R, Rabinowitz N, et al. Overcoming catastrophic forgetting in neural networks[J]. Proceedings of the national academy of sciences, 2017, 114(13): 3521-3526.
> >
> > [2] Zhang, Shizhou, et al. "Revisiting Generative Replay for Class Incremental Object Detection." Proceedings of the Computer Vision and Pattern Recognition Conference. 2025.
> >
> > [3] Wang, Yanjie, et al. "High-dimension Prototype is a Better Incremental Object Detection Learner." The Thirteenth International Conference on Learning Representations.
> >
> > [4] Mo, Qijie, et al. "Bridge past and future: Overcoming information asymmetry in incremental object detection." European Conference on Computer Vision. Cham: Springer Nature Switzerland, 2024.

---

> ### Author Response · Authors · 2025-11-21
>
> We sincerely thank the reviewer for the update and for raising the score. We appreciate the constructive feedback and the very helpful suggestions. Good luck to you as well.

---

### Official Review · Reviewer_V8s4 · 2025-11-02

**Soundness:** 3
**Presentation:** 2
**Contribution:** 3
**Rating:** 6
**Confidence:** 4

**Summary:**

The authors propose IIKC, a two-part framework for Incremental Object Detection (IOD): (1) Interference-Knowledge-Isolated Elastic Weight Consolidation leverages the old model’s mispredictions on new-task data to eliminate interference caused and rebuild the Bayesian posterior and parameter importance; and (2) Prototype-based Knowledge Calibration applies a learnable linear projection to compensate for semantic drift of old-class prototypes and then jointly retrains the classification head with current features. The approach outperforms strong baselines—covering both no-rehearsal and small-exemplar rehearsal regimes—across multiple stepwise and multi-step protocols on PASCAL VOC and MS-COCO.

**Strengths:**

1. The proposed IKI-EWC internalizes the IOD-specific “future-class- background” interference into computable posterior correction and importance fusion, yielding an end-to-end implementable path for conflict isolation. Compared with heuristic reweighting or soft masking, it offers an explicit probabilistic formulation with closed-form solutions (Eqs. 10 and 12).Which provides a systematic extension of EWC (new decomposition, a quantified $k$ , and a new fused importance $\tilde{I}$ ) that is both theoretically grounded and engineering-ready.

2. PKC addresses semantic drift using a lightweight projection together with prototype-based retraining, with small overhead and yielding clear gains (as confirmed by ablations). Moreover, compared with certain dual-teacher distillation schemes, the proposed method is simpler in both computation and implementation.

3. The paper is well structured and clearly written. Figures 1 and 2 intuitively illustrate the core idea and overall framework, enabling rapid understanding. It also provides an overall framework diagram and complete training pseudocode—explicitly specifying the input parameters γ, Top-K, and λ—which facilitates re-implementation and comparative ablation studies.

4. The evaluation experiments are thorough and in-depth, the results show consistent gains over strong baselines on VOC/COCO across stepwise/multi-step protocols. The authors analyze show low sensitivity to γ and that computed k beats extreme settings, and report runtime/memory costs.

**Weaknesses:**

1. The core of IKI-EWC is to accurately estimate interference regions, a procedure that relies entirely on pseudo-labels generated by the previous model ($M_{t-1}$ ) on the new data. If $M_{t-1}$ has degraded in performance or exhibits prediction bias, the interference estimation may be inaccurate, thereby affecting the overall performance of the framework. Although the paper notes this in the limitations section, it does not experimentally analyze how sensitive the method is to pseudo-label quality as a function of  $M_{t-1}$ performance.

2. The statistical robustness of the relative mass $k$ is not demonstrated. Since $k$  is computed as a ratio of proposal counts, it is highly sensitive to the IoU threshold, the number of proposals, class long-tail effects, and scale distributions. When sample size or the positive/negative balance fluctuates across stages, the variance of  $k$  can become large, causing the importance III to oscillate excessively. The paper lacks a systematic report of confidence intervals and sensitivity analysis for $k$ .

3. Evidence for the “approximate independence/orthogonality” assumption is limited. The derivations assume that “clean historical data” and “current data” are approximately independent, and cite an early-training gradient angle of ≈90° as support. However, orthogonality is not independence, and early mini-batches do not characterize the entire training trajectory. If representations become increasingly coupled later on—especially in multi-object/multi-scale settings—the premise underlying the posterior reconstruction weakens, potentially biasing the direction of the regularization. This point requires more extensive discussion.

4. PKC’s “prototype + linear projection” design is simple; however, compared with EFC’s anisotropic constraints with Gaussian-prototype updates, and LDC’s label-free learnable drift compensation, it might show clear shortcomings in the granularity of drift modeling, robustness and statistical sufficiency.

5. Several implementation details are insufficiently specified—for example, the confidence threshold for pseudo-labels, the method and computational cost of Hessian/Fisher estimation, and the sampling procedure used in PKC. In addition, it is not fully transparent whether data augmentation and preprocessing are exactly matched to the strong baselines in both the rehearsal-free and rehearsal-based regimes.

[1] Elastic Feature Consolidation for Cold Start Exemplar-Free Incremental Learning, ICLR 2024.
[2] Exemplar-free Continual Representation Learning via Learnable Drift Compensation, ECCV 2024.

**Questions:**

See above.

---

> ### Author Response · Authors · 2025-11-19
>
> We sincerely thank the reviewer for the helpful suggestions and for recognizing our work.
>
> **W1:**
> Thank you for the helpful suggestion. Following prior pseudo-label–based works [1][2][3], which are themselves sensitive to pseudo-label quality, we use a unified and strong pseudo-label configuration across all experiments to control this factor and enable fair comparison.
> To probe sensitivity to the quality of pseudo labels, we used three undertrained previous detectors at epochs 2, 4, and 6 on VOC 10 to 10. The row **base** reports the first task performance of that old model. The row **k** reports the estimated interference ratio under that model. The row **baseline** applies a common pseudo labeling strategy from IOD. The row **IKI-EWC** reports our method under the same setting.
>
> |                | epoch 2 | epoch 4 | epoch 6 |
> |----------------|---------|---------|---------|
> | base           | 49.7    | 69.7    | 72.8    |
> | $k$              | 0.0013  | 0.0025  | 0.0021  |
> | baseline       | 55.2    | 69.2    | 71.4    |
> | IKI-EWC        | 57.1 (+1.9) | 70.8 (+1.6) | 72.5 (+1.1) |
>
> Earlier checkpoints yield pseudo labels with lower precision and recall than later ones. These results indicate that IKI-EWC remains stable when pseudo labels are noisy and consistently outperforms the baseline at matched noise levels.
>
> We further vary the pseudo-label confidence threshold when computing IKI-EWC to simulate different pseudo-label qualities.
>
> **VOC 10-10:**
>
> | Confidence threshold | 1-20 (mAP) |
> |----------------------|-------------|
> | 0.20                 | 74.4 (+0.6) |
> | 0.50                 | 74.7 (+0.9) |
> | 0.75                 | 75.1 (+1.3) |
>
> Lower thresholds increase recall but reduce precision and introduce more noisy positives. Even at a low threshold of 0.2 where pseudo labels are very noisy, the performance of our method remains above the baseline of 73.8 mAP.
>
> **W2:**
> We thank the reviewer for the thoughtful comment. We acknowledge that the relative mass $k$, as a ratio of proposal counts, is subject to statistical variance and can be influenced by IoU thresholds, the proposal budget, class imbalance, and scale. In our framework, $k$ acts as a scaling factor that balances the two importance terms $I\_1$ and $I\_2$ in the derived posterior of Eq. 12.
>
> **Eq 12:**
> $\widetilde{\boldsymbol{I}\_{i}} = \frac{\boldsymbol{I}\_{1,i}\boldsymbol{I}\_{2,i}}{(1+k)^2 \boldsymbol{I}\_{2,i} + k^2 \boldsymbol{I}\_{1,i}}$
>
> To assess the impact of $k$ variation on performance, we scale $k$ by constants and report 1–20 mAP on VOC 10–10. The results show stable performance for broad ranges around the nominal value, with only mild degradation at extreme scales.
>
> |  $0.0\times k$  |  $0.5\times k$  | $1.0\times k$ | $5\times k$ | $100\times k$ | $1e8\times k$ |
> |-------------------|--------------|------------|------------|------------|------------|
> |74.5  |  74.7  |  75.1  |  74.9  |  74.5  |  74.3  |
>
> These results indicate that the method is robust to moderate fluctuations in $k$. In the extreme case $0.0\times k$, our formulation reduces to standard EWC. (Because the proposals used to compute $k$ are the same proposals used to estimate $I\_1$ and $I\_2$, the factors that perturb $k$ jointly affect $I\_1$ and $I\_2$ as well. For this reason, we focus on end-to-end analysis rather than an isolated ``$k$-only'' analysis in our paper.)
>
> **W3:**
> We appreciate the reviewer’s concern and agree that orthogonality is not the same as independence. In practice, strict independence is rarely attainable or verifiable. Even in incremental classification, where EWC[4] is widely used, tasks share a common feature extractor and many classes are semantically related. For instance on CIFAR-100, aquatic mammals (e.g., beaver, dolphin, otter, seal, whale), flowers (orchid, poppy, rose, sunflower, tulip), trees (maple, oak, palm, pine, willow), and vehicles (bicycle, bus, motorcycle, pickup truck, train) exhibit natural groupings. EWC and related methods typically do not require or prove strict independence across such groups.
>
> For detection, we adopt near-orthogonality as a measurable proxy for low correlation in parameter updates on non-conflicting regions, which supports a local approximate-independence view. Empirically, this approximation appears adequate. On VOC 10–10, we report the mean and standard deviation of the gradient angle over the full trajectory and over the final 100 iterations:
>
> | Setting              | Mean angle | Std of angle |
> |----------------------|------------|--------------|
> | Full training        | 88.30°     | 11.44°       |
> | Last 100 iterations  | 93.22°     | 10.73°       |

---

> ### Author Response · Authors · 2025-11-19
>
> **W4:**
> We appreciate the reviewer’s concern. Unlike classification, a single image fed to a detector produces $N$ region-level features rather than one image-level feature. As a result, the region features from the old and new models do not form a one-to-one correspondence. Our PKC is designed for detection. It uses a learnable projection to model semantic changes between the Top $K$ same-class region feature pairs ranked by $L_2$ distance from the two models, and then compensates the prototypes accordingly.
>
> EFC and LDC are proposed for classification. EFC estimates semantic drift by taking the difference between old and new features and assigns it to each class. LDC learns a linear layer to model the change between features from the two models. Under the VOC 10–10 detection setting we implement these methods. The results are as follows:
>
> | Method | 10–10 (mAP) |
> |--------|-------|
> | LDC    | 74.1  |
> | EFC    | 74.0  |
> | PKC    | 74.3  |
>
> PKC achieves the best mAP among the three while keeping the detection-oriented. Unlike EFC and LDC, which are conceived for classification with one-to-one feature alignment, PKC operates on region features without one-to-one correspondence, selects Top-$K$ same-class pairs.
>
> **W5:**
> Thank you for pointing this out. We specify the following implementation details for transparency.
> **The confidence threshold for pseudo-labels:**
> We use an adaptive pseudo-label threshold in all task settings. We fit a Gaussian mixture model to the confidence scores from the old model and set the threshold to the mean plus one standard deviation.
> **The method of Fisher estimation:**
> We follow the standard EWC procedure to compute the Fisher estimation. The diagonal Fisher is estimated as the expectation of squared gradients of the log likelihood at $\theta^{*}_{t-1}$. The derivation is given in Appendix Eq. 32.
>
> **Eq 32:**
> $$\boldsymbol{H}\_i \approx -\mathbb{E}\_{(x,y) \sim \mathcal{D}\_{1:t-1}} \left[ \left( \frac{\partial}{\partial \boldsymbol{\theta}\_i} \log p(y|x, \boldsymbol{\theta})\bigg|\_{\boldsymbol{\theta}\_i = \boldsymbol{\theta}\_{t-1,i}^\*} \right)^2 \right],
> \boldsymbol{H}\_i^{\*} \approx -\mathbb{E}\_{(x,y) \sim \hat{\mathcal{R}}\_{1:t-1}^\*} \left[ \left( \frac{\partial}{\partial \boldsymbol{\theta}\_i} \log p(y|x, \boldsymbol{\theta})\bigg|\_{\boldsymbol{\theta}\_i = \boldsymbol{\theta}\_{t-1,i}^*} \right)^2 \right].$$
>
> **The cost on a single RTX 4090 under VOC 10–10:**
>
> | Operation                                   | Time      | GPU memory | CPU memory|
> |---------------------------------------------|-----------|------------|----------|
> | Pseudo labeling + $k$ computation + Fisher | 256 s | 19,108 MB  |313.78 MB  |
>
> The end to end training time for VOC 10–10 is **10440 s** in Appendix Table 13.
>
> **PKC sampling:** In each iteration and for all settings, we sample two positive prototypes per foreground class. Background prototypes are sampled so that the background to foreground count matches the foreground to background ratio in the new task data.
>
> **Data augmentation and preprocessing:** We match ABR and BPF. We apply resize, horizontal flip with probability 0.5, image normalization, and conversion to BGR format.
> **Our training details are provided in the code included in the supplementary material.**
>
>
> [1] Zhang, Shizhou, et al. "Revisiting Generative Replay for Class Incremental Object Detection." Proceedings of the Computer Vision and Pattern Recognition Conference. 2025.
> [2] Wang, Yanjie, et al. "High-dimension Prototype is a Better Incremental Object Detection Learner." The Thirteenth International Conference on Learning Representations.
> [3] Mo, Qijie, et al. "Bridge past and future: Overcoming information asymmetry in incremental object detection." European Conference on Computer.
> [4] Kirkpatrick J, Pascanu R, Rabinowitz N, et al. Overcoming catastrophic forgetting in neural networks[J]. Proceedings of the national academy of sciences, 2017, 114(13): 3521-3526.

---

> > ### Comment · Reviewer_V8s4 · 2025-11-25
> >
> > The rebuttal has fully addressed my concerns. I will increase my rating. Good work.

---

> > > ### Author Response · Authors · 2025-11-25
> > >
> > > We sincerely thank the reviewer for the positive feedback and for increasing the rating. We truly appreciate your time and constructive comments.

---

### Official Review · Reviewer_w6Bq · 2025-11-04

**Soundness:** 3
**Presentation:** 3
**Contribution:** 3
**Rating:** 6
**Confidence:** 3

**Summary:**

This paper tackles the interference of knowledge between past/present knowledges in Incremental dense prediction problems. To mitigate this problem, the authors conduct quantative modeling based on bayesian analysis and propose novel algorithms called IKI-EWC. Also, the authors propose prototype based classifier correction algorithm to prevent the drift at the classfier level

**Strengths:**

- The authors well analysed the existing problem and then conducted rigorous mathematical analysis on that. This makes the motivation of the proposed method very strong
- The paper is well written and easy to follow.
-This paper provides extensive experimental results that can empirically support the authors' claim as well

**Weaknesses:**

- Authors approximated non-interference knowledge only by using current data. I agree that this would be the best/practical way of doing it under this setting but I wonder does it actually approximate the goal well? Is there any way of computing ground truth goal even by using whole ground data?

- I wonder why the authors used parameter regularisation methods. It is generally know to be poor compared to distillation methods. Can we use the IKI concept for distillation based methods as well?

- Prototype based classifier retain is quite common concept in classification CIL methods. Is it entirely novel in dense prediction tasks?

- Doesn't not regarding IKI concept on PKC levels conflicts with the other module (IKI-EWC)?

- Although the motivation is quite strong, improvements look marginal, especially when each module is solely applied.

**Questions:**

See weaknesses

---

> ### Author Response · Authors · 2025-11-19
>
> We sincerely thank the reviewer for the constructive feedback and for acknowledging our contributions.
>
> **W1:**
> We thank the reviewer for recognizing that using current data is a practical choice under the IOD setting. We validate interference isolation with an oracle study on VOC (15–5). We assume perfect future labels for stages $1:t-1$ and construct the ideal non-conflicting set
>
> $\mathcal{R}'\_{1:t-1}=\mathcal{R}\_{1:t-1}\setminus \mathcal{R}^{*}\_{1:t-1},$
>
> which we refer to as Oracle-EWC. We compare this to IKI-EWC, which estimates $\hat{\mathcal{R}}^{*}\_{1:t-1}$ at stage \(t\) without revisiting past data. Appendix Table 8 shows that IKI-EWC closely matches Oracle-EWC, indicating that removing interference directly at stage $t$ is a sound approximation.
>
> It is reasonable to explicitly model and remove interference to task $t$ at stage $t$. The current ground truth provides reliable anchors, the estimation uses only available data, and the posterior correction acts where the conflict actually occurs.
>
> **VOC 15–5 (Table 8):**
>
> | Method      | 1–15 | 16–20 | 1–20 |
> |-------------|------|-------|------|
> | EWC         | 76.8 | 61.3  | 72.9 |
> | IKI-EWC     | 76.7 | 63.1  | 73.3 |
> | Oracle-EWC  | 76.8 | 63.5  | 73.5 |
>
> These results support that our approximation is accurate and practical even without access to past data.
>
> **W2:** We thank the reviewer for the question. Parameter regularization and distillation rely on different signals. Parameter regularization estimates old-task importance from past data in a MAP view and constrains updates in parameter space. Distillation aligns responses on current-task data, which in incremental detection is dominated by new classes and often contains few or unlabeled old classes. This imbalance makes response alignment noisy for old classes and can propagate teacher errors, while importance-based regularization remains informative even when old-class evidence is scarce.
>
> The IKI idea is MAP-based and reduces the importance of parameters that interfere with the new task, reconstructing the MAP factorization in parameter space. We simply apply the IKI concept by distilling only on non-conflicting RoIs. Under the VOC 10–10 setting, the results are shown below.
>
> | Method      | 1–10 | 11–20 | 1–20 |
> |-------------|------|-------|------|
> | IKI-KD      |  73.0  |  74.8 | 74.0 |
> | IKI-EWC     | 75.0 | 75.2  | 75.1 |
>
> **W3:** We appreciate the reviewer’s perspective. We acknowledge that prototype ideas are well established in classification CIL. Our contribution lies in a detection oriented instantiation rather than proposing prototypes themselves. In classification a single image produces one image level feature. In detection a single image produces many region level features, typically around 512, so features from the old and new models do not form a one to one correspondence. We therefore select the Top K region features of the same class by ($L\_2$) distance and learn a trainable projection to model their semantic drift. We use this projection to compensate the prototypes and then retrain the classification head.
>
> **W4:** Thank you for pointing this out. The two modules do not conflict but are complementary and operate in different spaces under the principle of interference isolation. IKI-EWC applies MAP with a Laplace approximation in the parameter space of the region feature extractor and down-weights parameters aligned with interference (Eq. 12). PKC acts in the feature space after the region extractor and performs semantic drift compensation only on non-conflicting region features, then retrains the classifier with calibrated prototypes (Eq. 14). Because PKC operates on non-conflicting regions, it does not offset the suppression enforced by IKI-EWC.
>
> **Eq 11:**
> $\widetilde{\boldsymbol{I}\_{i}} = \frac{\boldsymbol{I}\_{1,i}\boldsymbol{I}\_{2,i}}{(1+k)^2 \boldsymbol{I}\_{2,i} + k^2 \boldsymbol{I}\_{1,i}}.$
>
> **Eq 14:**
> $\mathcal{L}^\text{proj} = \sum\_{i \in {\mathrm{TopK}}} \left\| \varphi(\mathbf{f}\_{t-1,i}) - \mathbf{f}\_{t,i} \right\|\_2^2.$

---

> ### Author Response · Authors · 2025-11-19
>
> **W5:**
> | Method  | VOC 10-10 | VOC 10-5 |
> |---------|----------|----------|
> | baseline | 73.8    |66.4|
> | IKI-EWC | 75.1 (+1.3)| 70.9 (+4.5)|
> | PKC     | 74.3  (+0.5)|67.9 (+1.5)|
>
> Thank you for the detailed question.
> We would like to clarify that in object detection, especially on mature benchmarks such as PASCAL VOC and COCO, such a gain is widely regarded as substantial in mAP. Compared with recent IOD papers [1][2], our improvements are also competitive.
> To further demonstrate effectiveness, we evaluated IKI-EWC and PKC with 20 independent random seeds under the VOC (10–10) setting. The results are shown below.
>
> **VOC 10–10:**
> | Method    |      Mean mAP     |     Std    | Two-sided p-value |
> |---------------------|------------------------|------------------|-------------------|
> | IKI-EWC |    75.1    | 0.16  | $4.15\times10^{-19}$ |
> | PKC        |      74.3    | 0.05  | $8.85\times10^{-20}$ |
>
> These results indicate consistent and statistically significant gains for each module when applied on its own. We compute p-values via a two-sided paired t-test on per-seed mAP improvements over the baseline, testing whether the mean gain is zero. A very small p-value means the observed improvement is unlikely to be due to random seed variation, supporting that our gains are not marginal. In our case the p-values are far below 0.001, indicating highly significant improvements.
>
> [1] Zhang, Shizhou, et al. "Revisiting Generative Replay for Class Incremental Object Detection." Proceedings of the Computer Vision and Pattern Recognition Conference. 2025.
>
> [2] Wang, Yanjie, et al. "High-dimension Prototype is a Better Incremental Object Detection Learner." The Thirteenth International Conference on Learning Representations.

---

> > ### Comment · Reviewer_w6Bq · 2025-11-26
> >
> > Thank you for the detailed rebuttal provided by the authors. As my concerns are mostly addressed, I will keep my positive scores for this paper.

---

> > > ### Author Response · Authors · 2025-11-26
> > >
> > > We sincerely thank the reviewer for the encouraging feedback and for maintaining the positive scores. We greatly appreciate your time and constructive comments.

---

### Author Response · Authors · 2025-12-01
**Summary comment (Part 3 of 3)**

[1] Wang, Yanjie, et al. "High-dimension Prototype is a Better Incremental Object Detection Learner." The Thirteenth International Conference on Learning Representations.

[2] Mo, Qijie, et al. "Bridge past and future: Overcoming information asymmetry in incremental object detection." European Conference on Computer.

[3] Zhang, Shizhou, et al. "Revisiting Generative Replay for Class Incremental Object Detection." Proceedings of the Computer Vision and Pattern Recognition Conference. 2025.

[4] Kirkpatrick J, Pascanu R, Rabinowitz N, et al. Overcoming catastrophic forgetting in neural networks[J]. Proceedings of the national academy of sciences, 2017, 114(13): 3521-3526.

[5] Elastic Feature Consolidation for Cold Start Exemplar-Free Incremental Learning, ICLR 2024.

[6] Exemplar-free Continual Representation Learning via Learnable Drift Compensation, ECCV 2024.

---

### Author Response · Authors · 2025-12-01
**Summary comment (Part 2 of 3)**

### Reviewer-wise concerns, our responses, and final reviewer comments:

After our detailed rebuttal, Reviewers `V8s4`, `Zo9q`, and `a9JU` **raised their scores**, and Reviewer `w6Bq` **kept a positive score**.
Below we summarize, for each reviewer, the main concerns we addressed and the final comments.

#### Reviewer `w6Bq`

- **Main concerns.**
  Reviewer `w6Bq` asked for an oracle-style evaluation with full ground-truth data, clarification of how the IKI concept extends to distillation-based methods, a clearer role of PKC in dense prediction and its relation to IKI-EWC, and whether per-module gains are significant.

- **Our responses.**
  We added an oracle comparison in Appendix Table 8 (`W1`) and showed that IKI-EWC closely matches this oracle. We implemented an IKI-style distillation variant on non-conflicting RoIs (`W2`) and clarified that PKC is a detection-specific, region-level prototype module that complements IKI-EWC (`W3`, `W4`). We also highlighted that gains of 4.5 and 1.5 mAP on VOC (10–5) are substantial compared to recent IOD works [1,2,3], and we confirmed robustness with 20 random seeds on VOC (10–10) (`W5`).

**Final comment from `w6Bq` (26 Nov 2025, 09:48)**

> "Thank you for the detailed rebuttal provided by the authors. As my concerns are mostly addressed, I will keep my positive scores for this paper."

#### Reviewer `V8s4`

- **Main concerns.**
  Reviewer `V8s4` requested sensitivity analyses for pseudo-label quality and the relative mass $k$, a deeper discussion of the approximate independence assumption, comparisons between PKC and EFC [5] / LDC [6], and more implementation details.

- **Our responses.**
  We used a unified, strong pseudo-label configuration following prior work [2,3], and added experiments with under-trained teachers and different confidence thresholds in `W1`, showing that IKI-EWC is stable under noisy pseudo-labels. We scaled $k$ and reported its impact on VOC (10–10), explaining why an end-to-end view is more appropriate than a "$k$-only" analysis (`W2`). We reported gradient-angle statistics and used CIFAR-100 to justify the approximate independence assumption (`W3`), implemented EFC[5] and LDC[6] under VOC (10–10) to show PKC compares favorably (`W4`), and added further implementation details and pointers to code in `W5`.

**Final comment from `V8s4` (25 Nov 2025, 10:12)**

> "The rebuttal has fully addressed my concerns. I will increase my rating. Good work."

#### Reviewer `Zo9q`

- **Main concerns.**
  Reviewer `Zo9q` asked for results in realistic settings with inaccurate pseudo-labels, clarification of computational complexity and scalability, and more detailed explanation of the method and derivations.

- **Our responses.**
  We reused the noisy pseudo-label experiments designed for Reviewer `V8s4` and, in `W1`, reported results with under-trained teachers and varied thresholds, showing that IKI-EWC remains robust. We reported training overhead and runtime in Appendix Table 13 and noted that Tables 2 and 3 already cover multi-task and large-scale setups, while additional analysis in `W2` and `Q2` shows that the extra cost is small and the method scales well. In our Official Comment (`W3`, `Q1`, `Q3`, `Q4`), we expanded the derivations, assumptions, and design rationale.

**Updated Official Review from `Zo9q` (modified: 20 Nov 2025, 23:24)**

> "UPDATES: After going through the delineated responses, I am raising my score. Good luck :)"

#### Reviewer `a9JU`

- **Main concerns.**
  Reviewer `a9JU` requested stronger comparison with state-of-the-art IOD methods, a clearer presentation of the IKI-EWC formulation, and explicit reporting of memory usage.

- **Our responses.**
  In `W1` and `Q1`, we added direct comparisons with GMDP-ABR [1] and RGR [3], showing that our method matches or exceeds them on VOC and COCO while avoiding generator training and extra storage, and we discussed suitability for low-complexity, memory-constrained regimes. In `W2` and `Q2`, we clarified the IKI-EWC derivation and assumptions. In `W3` and `Q3`, we compared memory usage with GMDP-ABR [1] and showed that our additional cost is much smaller.

**Final comment from `a9JU` (27 Nov 2025, 11:10)**

> "The author's response resolved my concerns, and I raise my score."

---

### Author Response · Authors · 2025-12-01
**Summary comment (Part 1 of 3)**

### Declaration about the OpenReview bug:

We did not use, propagate, or benefit from the OpenReview bug that exposed the identities of authors, reviewers, and area chairs.

Our last interactions with the four reviewers ended at the following times, together with their initial and final scores:

* Reviewer `w6Bq`: **`26 Nov 2025, 09:48`**
  * Initial score: **6** → Final score: **6**

* Reviewer `V8s4`: **`25 Nov 2025, 10:12`**
  * Initial score: **6** → Final score: **8**

* Reviewer `Zo9q`: **`20 Nov 2025, 23:24`**
  * Initial score: **4** → Final score: **6**

* Reviewer `a9JU`: **`27 Nov 2025, 11:10`**
  * Initial score: **6** → Final score: **8**

---

### Overall strengths highlighted by reviewers:

Across all four reviews, the paper is consistently recognized for novelty, method design, empirical strength, and clarity and reproducibility. We summarize these strengths as follows:

* **Novelty and theoretical contribution.**
  Reviewer `w6Bq` (`Strength 1`), Reviewer `V8s4` (`Strength 1`), Reviewer `Zo9q` (`Strengths 1 and 2`), and Reviewer `a9JU` (`Strength 1`) all highlight the novelty of our approach. They note that the paper provides a rigorous analysis of the "future-class as background" problem. They also emphasize that the Bayesian reformulation of EWC[4] and the dual-module framework (IKI-EWC + PKC) offer a principled and meaningful advance for incremental object detection.

* **Method design and practicality.**
  Reviewer `V8s4` (`Strengths 1 and 2`), Reviewer `Zo9q` (`Strength 2`), and Reviewer `a9JU` (`Strength 1`) praise the overall design of the method. They view IKI-EWC as an end-to-end, implementable mechanism for conflict isolation. They also describe PKC as a lightweight, detection-oriented calibration module that preserves old knowledge without storing past images and is simple to deploy in practice.

* **Empirical evaluation and robustness.**
  Reviewer `w6Bq` (`Strength 3`), Reviewer `V8s4` (`Strength 4`), and Reviewer `a9JU` (`Strength 2`) agree that the empirical study is strong. They note that experiments on VOC and COCO are broad and in-depth, and that our method shows consistent gains over strong baselines under multiple IOD protocols. They also recognize that the ablations and sensitivity analyses support the robustness of both IKI-EWC and PKC.

* **Clarity and reproducibility.**
  Reviewer `w6Bq` (`Strength 2`) and Reviewer `V8s4` (`Strength 3`) comment that the paper is well written and well structured. They highlight that the figures, framework diagram, and training pseudocode with explicit hyperparameters make the method easy to understand, re-implement, and compare against in future studies.

---

### Meta-Review · Area_Chair_qYU7 · 2026-01-07

**Summary:**

This paper receives 3x marginal accept and 1x marginal reject. All reviewers raised their scores after the rebuttal and discussion phases, and thus making it a clear accept. The strengths are: 1) the paper is solving an important and practical problem of incremental detection. 2) well analyzed the existing problem and then conducted rigorous mathematical analysis; 3) paper is well written and easy to follow; 4) Experiments on PASCAL VOC and MS-COCO are broad, and ablations show both modules are useful. The major weaknesses pointed out by the reviewer who gave a marginal reject are: 1) Dependence on Pseudo-Labels and Sensitivity to Noise; 2) Computational Complexity and Scalability Concerns; 3) Residual interference. Nonetheless, these concerns are well-addressed by the authors in the rebuttal and thus leading to a raise in the score to positive. The questions from other reviewers are also well answered in the rebuttal, which lead to further increase of the scores. The AC follows the recommendation of the reviewers to accept the paper.

**Reviewer Concerns:**

The major weaknesses pointed out by the reviewer who gave a marginal reject are: 1) Dependence on Pseudo-Labels and Sensitivity to Noise; 2) Computational Complexity and Scalability Concerns; 3) Residual interference. Nonetheless, these concerns are well-addressed by the authors in the rebuttal and thus leading to a raise in the score to positive. The questions from other reviewers are also well answered in the rebuttal, which lead to further increase of the scores.

**Reviewer Scores:**

It's clear from the rebuttal and discussions that the reviewers' doubts are well-addressed and the scores are changed to positive.

---

### Decision · Program_Chairs · 2026-01-26

Accept (Poster)